# The Impact of COVID-19 on Maternal Mental Health during Pregnancy: A Comparison between Canada and China within the CONCEPTION Cohort

**DOI:** 10.3390/ijerph191912386

**Published:** 2022-09-28

**Authors:** Nicolas Pagès, Jessica Gorgui, Chongjian Wang, Xian Wang, Jin-Ping Zhao, Vanina Tchuente, Anaïs Lacasse, Sylvana Côté, Suzanne King, Flory Muanda, Yves Mufike, Isabelle Boucoiran, Anne Monique Nuyt, Caroline Quach, Ema Ferreira, Padma Kaul, Brandace Winquist, Kieran J. O’Donnell, Sherif Eltonsy, Dan Chateau, Gillian Hanley, Tim Oberlander, Behrouz Kassai, Sabine Mainbourg, Sasha Bernatsky, Évelyne Vinet, Annie Brodeur-Doucet, Jackie Demers, Philippe Richebé, Valerie Zaphiratos, Anick Bérard

**Affiliations:** 1Research Center CHU Ste-Justine, Montreal, QC H3T 1C5, Canada; 2Faculty of Medicine, Université Claude Bernard Lyon 1, 69003 Lyon, France; 3Faculty of Pharmacy, University of Montreal, Montreal, QC H3T 1J4, Canada; 4College of Public Health, Zhengzhou University, Zhengzhou 450001, China; 5Health Sciences Department, Université du Québec en Abitibi-Témiscamingue, Rouyn-Noranda, QC J9X 5E4, Canada; 6Faculty of Medicine, School of Public Health, University of Montreal, Montreal, QC H3T 1J4, Canada; 7Faculty of Medicine, McGill University, Montreal, QC H3G 2M1, Canada; 8Department of Epidemiology & Biostatistics, Western University, London, ON N6A 5W9, Canada; 9ICES Western, Western University, London, ON N6A 5W9, Canada; 10Department of Family Medicine, Protestant University in Congo, Kinshasa II, Kinshasa P.O. Box 4745, Democratic Republic of the Congo; 11Department of Obstetrics and Gynecology, School of Public Health, University of Montreal, Montreal, QC H3N 1X9, Canada; 12Department of Pediatrics, CHU Sainte-Justine, University of Montreal, Montreal, QC H3T 1C5, Canada; 13Department of Microbiology, Infectious Diseases and Immunology, University of Montreal, Montreal, QC H3T 1J4, Canada; 14Pharmacy Department, CHU Sainte-Justine, Montreal, QC H3T 1C5, Canada; 15Department of Medicine, 4-120 Katz Group Centre for Pharmacy and Health Research, University of Alberta, Edmonton, AL T6G 2R7, Canada; 16Department of Community Health and Epidemiology, College of Medicine, University of Saskatchewan, Saskatoon, SK S7N 5E5, Canada; 17Yale Child Study Center, Department of OB/GYN and Reproductive Sciences, Yale School of Medicine, New Haven, CT 06510, USA; 18Douglas Research Center, Department of Psychiatry, McGill University, Montreal, QC H4H 1R3, Canada; 19Rady Faculty of Health Sciences, College of Pharmacy, University of Manitoba, Winnipeg, MB R3E 0W2, Canada; 20Manitoba Center for Health Policy, Winnipeg, MB R3E 3P5, Canada; 21Department of Obstetrics & Gynaecology, University of British Columbia, Vancouver, BC V6T 1Z4, Canada; 22Department of Pediatrics, School of Population and Public Health, University of BC, Vancouver, BC V6T 1Z4, Canada; 23Department of Clinical Epidemiology, UMR 5558 CNRS, Clinical Investigation Centre, Inserm-Hospices Civils de Lyon, Claude Bernard University Lyon 1, 69003 Lyon, France; 24Divisions of Clinical Epidemiology and Rheumatology, McGill University Health Centre, Montreal, QC H3A 0G4, Canada; 25Dispensaire Diététique de Montréal/Montreal Diet Dispensary, Montreal, QC H3H 1J3, Canada; 26Department of Anesthesiology and Pain Medicine, CIUSSS de l’Est de l’Ile de Montreal, Maisonneuve-Rosemont Hospital, University of Montreal, Montreal, QC H1T 2M4, Canada

**Keywords:** COVID-19 pandemic, maternal mental health, pregnancy, Edinburgh Perinatal Depression Scale (EPDS), Generalized Anxiety Disorders (GAD-7), stress

## Abstract

The effect of the COVID-19 pandemic on maternal mental health has been described in Canada and China but no study has compared the two countries using the same standardized and validated instruments. In this study, we aimed to evaluate and compare the impact of COVID-19 public health policies on maternal mental health between Canada and China, as we hypothesize that geographical factors and different COVID-19 policies are likely to influence maternal mental health. Pregnant persons >18 years old were recruited in Canada and China using a web-based strategy. All participants recruited between 26 June 2020 and 16 February 2021 were analyzed. Self-reported data included sociodemographic variables, COVID-19 experience and maternal mental health assessments (Edinburgh Perinatal Depression Scale (EPDS), Generalized Anxiety Disorders (GAD-7) scale, stress and satisfaction with life). Analyses were stratified by recruitment cohort, namely: Canada 1 (26 June 2020–10 October 2020), Canada 2 and China (11 October 2020–16 February 2021). Overall, 2423 participants were recruited, with 1804 participants within Canada 1, 135 within Canada 2 and 484 in China. The mean EDPS scores were 8.1 (SD, 5.1) in Canada 1, 8.1 (SD, 5.2) in Canada 2 and 7.7 (SD, 4.9) in China (*p*-value Canada 2/China: *p* = 0.005). The mean GAD-7 scores were 2.6 (SD, 2.9) in China, 4.3 (SD, 3.8) in Canada 1 (*p* < 0.001) and 5.8 (SD, 5.2) in Canada 2 (*p* < 0.001). When adjusting for stress and anxiety, being part of the Chinese cohort significantly increased the chances of having maternal depression by over threefold (adjusted OR 3.20, 95%CI 1.77–5.78). Canadian and Chinese participants reported depressive scores nearly double those of other crises and non-pandemic periods. Lockdowns and reopening periods have an important impact on levels of depression and anxiety among pregnant persons.

## 1. Introduction

For almost three years, the COVID-19 pandemic has greatly impacted individuals worldwide. The world watched China lock down the border of the provinces of Wuhan and Hubei rapidly in order to contain COVID-19 on 23 January 2020. Shortly after, on 11 March 2020, the World Health Organization declared COVID-19 a pandemic that would soon ravage the entire world [1]. Governments worldwide reacted to the threat of the pandemic in different ways, with public health measures ranging from regional lockdowns and stay-at-home orders to extended quarantines, curfews, restrictions of services in private and public sectors, and the shutdown of entire industries, among others [2,3]. These measures, along with the uncertainty that the pandemic brought on, may have intensified emotional distress, especially among pregnant persons [4,5]. Indeed, it has been observed that the crisis has severely affected people’s mental health, leading to a prevalence of depression ranging from 8.3% to 48.3% [6,7,8].

Exposure to crises and stressful events during pregnancy is known to have long-term effects on the mental health of pregnant individuals as well as on the neuronal development of their offspring [9,10,11,12]. Indeed, in January 1998, an ice storm crisis in the Canadian provinces of Québec and Ontario resulted in power losses for nearly 3 million people for as long as 40 days. Project Ice Strom by Laplante et al. [13] demonstrated that prenatal exposure to a stressful natural disaster was associated with lower cognitive and language abilities in 2-year-old children [14,15]. Additionally, maternal stress and anxiety during pregnancy are also known to be associated with some maternal outcomes such as low birth weight and prematurity [16,17].

To investigate the impact of the COVID-19 pandemic on maternal and perinatal outcomes, the CONCEPTION cohort was initiated by Bérard et al. [18]. Pregnant persons were included worldwide, but Chinese and Canadian participants were among the most represented. The effect of the pandemic in both countries has been well-described individually, showing an increased risk of depression and anxiety in pregnant persons that may lead to short and long-term impacts on mothers and children [5,19,20,21,22,23,24]. To the best of our knowledge, the only study comparing the impact of COVID-19 on pregnant persons between China and Canada, among other countries, is a meta-analysis by Ghazanfarpour et al. conducted in October 2020 [25]. They pooled a number of studies across countries, and though they had highly heterogenous results, they found that the pooled prevalence of anxiety was 56% in Canada and 0.3–29% in China, while the pooled prevalence of depression was 37% in Canada and 11–29% in China [25]. Despite the lack of result generalizability and heterogeneity of the pooled studies, specifically in relation to the tools used to assess anxiety and depression, they concluded that COVID-19 imposed increased pressure on the emotional well-being of expectant mothers due to the fear of infection, infecting those around them, restricted access to healthcare during their pregnancy, and overall restrictions on their daily activities [25]. Lok et al. aimed to investigate fear and childbirth experience during both the pregnancy and postpartum periods in the COVID-19 crisis, recruiting Canadian and Chinese participants; however, the study is still ongoing [26]. In the general population, Lee et al. performed a meta-analysis where they compared depression outcomes across country in relation to the response time of each government [2]. Of note, the prevalence of depressive symptoms was 21.4% across the 33 included countries [2]. They found that early on in the pandemic (before December 2020), governments that acted promptly to reduce the spread of COVID-19 improved not only the physical but also the mental health of their population [2]. However, given that studies were performed rapidly and on small scales early on in the pandemic, the heterogeneity does not allow for direct and robust comparisons to be made between countries [25]. As we will continue to evaluate the impact of public health measures and the pandemic itself on maternal and perinatal outcomes, it is of utmost importance to generate scientific findings using validated tools to measure maternal mental health outcomes specifically.

The prevalence of COVID-19 and associated public health measures implemented to contain the pandemic differed between Canada and China, as described in Ghazanfarpour et al. [25]. China was the first country affected by COVID-19 at the end of December 2019. On the one hand, the Chinese government rapidly implemented a zero COVID strategy in hopes of eradicating the virus from the country [27,28]. To achieve this, they implemented strict lockdowns and cross-provincial channel barriers and forbade individual gatherings starting in January 2020 [29,30]. Measures were then eased on April 2020; however, the zero COVID strategy remained active with a precise system of prevention and controls, including the mass testing and isolation of positive cases in quarantine centers [27,28,31]. At the same time, there was restricted access to hospital centers for routine visits or follow-up of chronic illnesses, including obstetrical follow-ups [27,31,32,33]. On the other hand, Canada attempted to mitigate the spread of the virus and associated hospitalizations/deaths while also balancing economic activities. However, the zero COVID-19 approach was not considered in most Canadian provinces (i.e., Alberta, British Columbia, Manitoba, Ontario, Quebec and Saskatchewan) and rather used only in the territories (i.e., Northwest Territories, Nunavut, and Yukon) as well as the Maritimes (i.e., New Brunswick, Newfoundland and Labrador, and Prince Edward Island) [34]. Through Canadian public health mitigation strategies, most Canadian provinces went through two periods of severe COVID-19 restrictions to limit the spread of the virus (in spring 2020 and fall/winter 2020), with extended lockdowns, curfews, and no individual gatherings [32,34]. 

While there are stark differences in the way that the COVID-19 pandemic was handled in China and Canada, it is important to note some similarities when it comes to more general public health strategies. First, maternal healthcare has been made a priority in both countries. Indeed, in the last two decades, China has made reducing maternal and under-5 mortality one of its priorities and now has a similar maternal mortality rate (18.3 per 100,000 livebirths in 2018) to that of Canada (8.4 per 100,000 livebirths in 2020) and the Western world (12.0 per 100,000 livebirths in 2017) [35,36,37,38]. China implemented the Five Strategies for Maternal and Newborn Safety and the Healthy China initiative in 2016 [36,39], while in Canada the Program for Prevention of Maternal Morbidity and Mortality was launched in 2016 [40]. When looking specifically at the medical follow-up during pregnancy, both countries recommend several examinations such as blood tests, an oral glucose tolerance test, HIV screening, as well as regular physical and ultrasound examinations [41,42,43,44,45]. Moreover, recommendations regarding smoking and the use of alcohol during pregnancy are similar in Canada and China [41,44]. The Chinese Medical Association recommends 7 to 11 antenatal care hospital visits [45], and in Canada, 2 to 3 antenatal hospital visits are recommended along with the regular follow-up by a family physician or obstetrician, which adds up to the same number of follow-ups overall [46]. Both recommend an increased number of visits for pregnant persons at risk and towards the end of pregnancy [45,46]. In terms of access to this antenatal follow-up, Chinese persons have the choice between universal health coverage which occurs in public institutions and are covered by social security contributions, or the private/international institutions which typically entails the subscription to a health insurance, either paying for the totality of healthcare services or a portion of it [35,47,48]. Canadian persons on the other hand have universal access for all their healthcare services, funded through tax-payers, available to them [49]. This said, when it comes to COVID-19, both countries limited onsite hospital visits for pregnant persons and promoted online counseling and training programs during lockdowns [32,33]. Differences in cultural habits and beliefs during pregnancy can be observed. In the Chinese culture, the pregnant person is considered vulnerable and requires continued rest [50]. As a consequence, Chinese pregnant individuals exist the work force early on in their pregnancy compared to Canadians [50]. Additionally, traditional taboos in China, such as “not walking too fast” to avoid spontaneous miscarriage and restriction on certain type of food or activities can make pregnancy a different experience in the two countries [50,51].

The COVID-19 pandemic generated uncertainty since January 2020, which has been directly correlated with increased stress in the general and pregnant populations [5,6,7,9,19,20,21,24,25]. This stress is the result of limited accurate information on the virus, limited availability of proven therapeutics, contradicting vaccination campaigns, and strong public policies to reduce the transmission of the virus [27,28,31,52]. For example, Lebel et al. surveyed pregnant individuals at the beginning of the pandemic in Canada and reported elevated clinically relevant symptoms of depression (37%) and anxiety (57%) comparted to similar pre-pandemic scores [5]. Moreover, the implementation of COVID-19 measures in the long run could increase the prevalence of depressive and anxiety symptoms because of social isolation and limited access to social and medical support [2,25]. As such, based on the current state of the literature summarized herein, we hypothesized that the more restrictive and lasting the COVID-19 measures, the greater the impact on maternal mental health.

Therefore, considering Canada and China continue to deal with the pandemic in different ways but have overall similar maternal health policies and public health strategies regarding pregnancy, the CONCEPTION study gave us the opportunity to answer this knowledge gap on the direct impact, specifically assessed with validated and standardized instruments in both countries. This is currently lacking in the present literature. Indeed, knowing the numerous complications of maternal mental health on the delivery and the neuronal development of their offspring’s [9,10,11,14,16,17], and because the COVID-19 pandemic continues to affect the world, we aimed to evaluate and compare the impact of COVID-19 measures on maternal mental health in order to inform healthcare professionals and decision makers on the impact of such decisions, specifically among Canadian and Chinese pregnant persons.

## 2. Methods

### 2.1. Study Design

The CONCEPTION study has been described in detail in Bérard et al. [18]. The recruitment into the CONCEPTION cohort started on 26 June 2020 and is ongoing. The present analysis includes Chinese and Canadian pregnant persons recruited between 26 June 2020 and 16 February 2021. Specifically, Chinese participants were recruited between 11 October 2020 and 16 February 2021. During this period in Zhengzhou, where all Chinese participants were recruited in university-affiliated hospitals, there was no lockdown nor any COVID-19 specific restrictions. In our previous study within the CONCEPTION cohort, we determined that the waves of the pandemic had a differential impact on maternal mental health [18], as such, it was important to compare cohorts based on calendar time. As such, we separated our Canadian participants into two cohorts based on their time of recruitment, namely: Canada 1—participants were enrolled between 26 June 2020 and 10 October 2020, Canada 2—participants were enrolled between 11 October 2020 and 16 February 2021. Canada 1 captures the summer of reopening in 2020 following the end of the first wave, therefore comparable in terms of COVID-19 conditions and measures with the Chinese cohort, whereas Canada 2 captures the second and more restrictive lockdown period in Canada.

In Canada, the recruitment of pregnant persons was web-based, using regular daily posting on different social media platforms (Facebook, Twitter, Instagram, TikTok and LinkedIn). To reach as many pregnant persons as possible, the study was promoted by mother/child and pregnancy support groups (e.g., Facebook—Dr. MILK as well as privately run “mom” groups), established hashtag strategies, influencers on Facebook and Instagram with a substantial following, and through communication specialists affiliated with our team’s respective Canadian universities. Online recruitment using anonymized surveys has been used in several other similar studies [53,54,55,56]. Indeed, as we wanted to collect and access data in real-time and rapidly given the ever-evolving nature of the COVID-19 pandemic, the use of an easily accessible online questionnaire is pertinent. A web-based questionnaire was easy to use for participants as they could access it using any electronic device when most convenient for them, a strategy tailored for young adults, our target population. Additionally, because we wanted to reach as many pregnant persons as possible to ensure as much diversity and representation as possible, the web-based strategy was preferred. To further ensure that our sample would be representative, recruitment was also conducted in person at the Montreal Dietetics Dispensary, which provides support to low-income and newly arriving mothers. In China, recruitment was carried out in person in three central hospitals from the Henan Province by investigators when pregnant persons came to the hospital for their clinical follow-up and were handed an electronic device where they completed the same questionnaire online independently. The difference in recruitment strategy was due to restricted social media access in China [57].

All pregnant individuals aged 18 years or older and able to read one the following language (French, English, Spanish, Portuguese or Mandarin) were eligible. The study instrument was created in English, translated to all available languages, and back translated to English in order to ensure their validity. Individual consent was obtained from pregnant persons and data was collected online using the secure platform SurveyMonkey^®^. All data were then downloaded on a secure server of our hospital at CHU Ste-Justine, Montreal, Quebec. A complete version of the instrument is available in the online Appendix A.

### 2.2. Data Collection

The baseline questionnaire was tested on 10 English-speaking and 10 French-speaking pregnant persons to ensure that questions were understood the same way in the two main languages of use. This questionnaire took on average 25 min to complete.

We collected several variables through our study instrument on SurveyMonkey^®^. All following variables were self-reported by pregnant persons.

(A)We first collected general maternal characteristics and health history (past history and since the start of pregnancy) in order to define our study groups in detail. These variables include: (1) general and socio-demographic information: gestational age (continuous), maternal age (continuous), pre-pregnancy height and weight to calculate the body mass index (continuous), ethnicity (Aboriginal, Asian, Black, Caucasian/white, Hispanic, other), annual household income (categorized as <$30,000, $30,000–$60,000, $60,001–$90,000, $90,001–$120,000, $120,001–$150,000, $150,000–$180,000 and >$180,000), years of education (continuous), living situation (with a partner, parents or family, alone), area of residence (urban, rural, suburban), country of residence (Canada, China); (2) Health behaviors including sports, smoking, alcohol and drug use (yes/no); (3) Comorbidities and medication use, including medications available over the counter (OTC); (4) Work/employment status and changes in status following the onset of the COVID-19 crisis and (5) Present experiences related to the COVID-19 pandemic.(B)We collected data on COVID-19, to measure the positivity rate and familial impact of COVID-19 throughout the pandemic. These variables include: (1) COVID-19 testing (yes/no) and diagnosis by a positive test (yes/no) (2) Number of immediate or extended family member(s) and/or close friends tested positive for COVID-19.(C)We assessed the impact of the public health measures on the pregnancy experience and changes in birth plans related to the COVID-19 pandemic by collecting information on: (1) Support by primary prenatal care provider(s) and resources available, (2) Type of prenatal classes/information, (3) Support persons not permitted during delivery, (4) Family and friends not permitted in hospital, (5) Separation with newborns after delivery, (6) Concerns about breastfeeding, and (7) All concerns regarding changes in the birth plan and delivery related to COVID-19 were measured on a 4-category ordinal scale; possible responses were “not concerned at all”, “a little concerned”, “moderately concerned” and “very concerned”.(D)As a proxy for the hardships pregnant participants endured, we asked about the impact of the COVID-19 pandemic on the: (1) financial situation, (2) family income, (3) daily routine, (4) food access, (5) medical health care access excluding mental health, (6) mental health treatment access, (7) access to family, extended family, and non-family social supports, and (8) work situation. Those variables were measured on a 4-category ordinal scale; possible responses were “no change”, “mild”, “moderate” and “severe”.(E)We lastly assessed maternal mental health during the COVID-19 pandemic by measuring: (1) Maternal depression during the pandemic, using the Edinburgh Perinatal Depression Scale (EPDS) [58], (2) Anxiety during the pandemic, using the generalized anxiety disorders scale (GAD-7) [59], (3) Satisfaction with life, comparing the time prior to vs. since the start of the COVID-19 pandemic using a 4-category ordinal scale with responses ranging from “very satisfied” to “very unsatisfied” and (4) Stress due to this COVID-19 pandemic using a visual analog scale ranging from 0 (no stress) to 10 (maximum stress). We chose to use the EDPS and GAD-7 instruments to assess maternal mental health. The EDPS score has been validated in Mandarin, French and English [60,61,62]. This instrument is composed of 10 items. Each item poses a question and is scored from 0 to 3, and the total scores range from 0 to 30. With a cut-off value of ≥13 representing severe depression, this tool has a sensibility of 66% and a specificity of 95% for the screening of depression [63]. The GAD-7 scale has also been validated in Mandarin, English and French [64,65,66]. This instrument is comprised of 7 items, each item poses a question and is scored from 0 to 3 and the total score ranges from 0 to 21. With a cut-off value >9 representing moderate to severe anxiety, this score has a sensibility of 89% and a specificity of 82% for the screening of anxiety [67]. As such, we have categorized depression symptoms as continuous measure first and further classified as moderate to severe (if EPDS > 9) and severe (if EPDS ≥ 13) [22]. Similar to this, anxiety symptoms were classified as moderate to severe (if GAD-7 > 9), and severe (if GAD > 15) [23]. These cut-offs are determined by the tools themselves [22,23].

### 2.3. Data Analyses

Given that pandemic waves have had an impact on maternal depression in Canada [18] and given that the time of response from the governments when facing the pandemic has an impact on mental health in the general population [2], analyses were stratified according to the recruitment cohort (Canada 1, Canada 2, and China). As described above, each cohort was recruited at a specific time-period during the pandemic and mothers had therefore lived the pandemic differently. We first compared the mean maternal depression and anxiety scores (Canada 1 vs. China, Canada 2 vs. China) as well as the frequency of moderate to severe depression (EPDS > 9) and anxiety (GAD-7 > 9), and severe depression (EPDS ≥ 13) and anxiety (GAD-7 > 15). We also described maternal satisfaction with life before and after the start of the COVID-19 pandemic and mean of overall stress were also compared. We then evaluated COVID-19 testing in each cohort. Finally, we compared the pregnancy experiences, changes in birth plans by cohort and the impact of COVID-19 on financial situation and daily life. Depending on whether the variables were continuous or categorical, Student’s *t*-tests or chi-square statistics were used to compare means with standard deviations or percentages for all variables.

We quantified the determinants of depression (EPDS > 9) during pregnancy by quantifying crude and multivariate associations using logistic regression models, considering the cohort of recruitment, maternal anxiety (continuous), maternal stress (continuous), maternal age (continuous), pre-pregnancy body mass index (continuous), weeks’ gestation at recruitment (continuous), employment status (employed [reference], unemployed or on welfare), years of education (continuous), and annual household income (categorized as defined above) as predictor variables. These adjustment variables were determined *a priori* and are based on our previous findings within the CONCEPTION Cohort [18]. Odds ratios (ORs) and 95% confidence intervals (CIs) were calculated. Missing data are presented in the tables and figures for each variable. Given the study design and recruitment strategy, missing data were not considered in the analyses.

All statistical analyses were performed using Excel (version 16.60) and R Studio (version 4.1.0). The CHU Sainte-Justine’s Research Ethics Committee approved the study (no. MP-21-2021-2973), which authorized the recruitment of subjects in both countries.

## 3. Results

### 3.1. Description of Participants

A total of 2423 participants were recruited between the 24 June 2020 and the 16 February 2021, with, respectively 1804 participants in the Canada 1 cohort, 135 in the Canada 2 cohort, and 484 in China (Figure 1).

Pregnant persons were on average 31.6 years old (standard deviation [SD, 4.4]) but were significantly younger in China (30.2 [SD, 4.4], *p* < 0.001) (Table 1). Gestational age at recruitment also differed significantly among cohorts with a mean of 33.3 weeks of gestation (SD, 7.7) in the Chinese cohort vs. 24.7 (SD, 9.8) and 20.7 (SD, 9.5) weeks in the Canada 1 and Canada 2 cohorts, respectively (Table 1). Overall, 87.7% of Chinese participants were recruited in their 3rd trimester compared to 45.9% (*p* < 0.001) and 25.9% (*p* < 0.001) in the Canada 1 and Canada 2 cohorts, respectively (Table 1). Canadian participants were less likely to be cared for by obstetricians during their pregnancy than in China (Canada 1: 55%/Canada 2: 52.6% vs. China: 96% *p* < 0.001). Moreover, education level was higher among Canadian participants (Canada 1: 16.8 years of education [SD, 4.5], Canada 2: 15.8 [SD, 6.1] and China: 14.4 [SD, 3.1], *p* < 0.001) (Table 1). In terms of employment status, only 59.1% of Chinese participants were employed (*p* < 0.001 compared to Canada 1 and Canada 2) and 32.1% of them were not employed (vs 1.9% Canada 1 and 0.8% Canada 2, *p* < 0.001). Differences in area of residence were also noticed with most Chinese participants living in an urban area (88.1% vs. 40.9% Canada 1 and 53.1% Canada 2, *p* < 0.001) (Table 1). Finally, OTC medication use was more common in Canada during pregnancy with 67.1% (Canada 1) and 60.2% (Canada 2) of participants reported taking OTC medication while pregnant when only 5.4% of participants in China did (Appendix A). Pregnancy history data of pregnant participants is presented is Table 2. Alcohol and smoking habits as well as drug use and physical activity of included participants are presented in Appendix A.

### 3.2. Maternal Mental Health—Depression, Anxiety, Stress Level and Satisfaction with Life

The mean maternal depression score using the EDPS was 8.1 (SD, 5.2) with significant differences between cohorts, participants from the Canada 2 group having the highest EDPS score (10.5, SD, 5.9) compared to Canada 1 (8.1, SD, 5.2) and China (7.7, SD, 4.9) (*p*-value Canada 2/China: *p* = 0.005) (Figure 2a). The prevalence of moderate to severe depression (EPDS > 9) was lower in China (33.4%) when compared to the Canada 1 cohort (37.8%) (*p* = 0.92) and when compared to Canada 2 cohort (54.7%) (*p* < 0.001) (Figure 2b). Similarly, we observed the same trends for the prevalence of severe depressive symptoms (EPDS ≥ 13); it was lower in China (17.7%) compared to the Canada 1 cohort (21.6%) (*p* = 0.08) and compared to Canada 2 cohort (36.8%) (*p* < 0.001) (Figure 2c).

Maternal anxiety symptoms, measured with the GAD-7 scale, also varied across cohorts (Figure 3a). Indeed, the overall mean maternal anxiety score was 4.0 (SD, 3.8). The lowest maternal anxiety score was reported in China (2.6 [SD, 2.9]) when compared to the Canada 1 cohort (4.3 [SD, 3.8]) (*p* < 0.001) and when compared to Canada 2 cohort (5.8 [SD, 5.2]) (*p* < 0.001) (Figure 3a). The prevalence of moderate to severe anxiety (GAD-7 > 9) was lower in China (0.9%) compared to the Canada 1 cohort (8.9%) (*p* < 0.001) and the Canada 2 cohort (15.7%) (*p* < 0.001) (Figure 3b). Severe maternal anxiety symptoms are presented in Figure 3c. The prevalence of severe anxiety (GAD-7 > 15) was (2.5%) in the Canada 1 cohort and (10.4%) in the Canada 2 cohort. It is important to note that no Chinese participants had severe anxiety at the time of recruitment.

Regarding maternal overall stress level related to COVID-19, significant differences were also identified between Canada and China (Figure 4). The mean stress level in China was 2.51 (SD, 2.05) compared to 4.6 (SD, 2.1) in the Canada 1 cohort (*p* < 0.001) and 5.5 (SD, 2.0) in the Canada 2 cohort (*p* < 0.001) (Figure 4). Differences in the satisfaction with life before the pandemic and at the time of recruitment in each cohort are reported in Figure 5. Satisfaction with life prior the pandemic compared to when participants completed their questionnaire differed significantly in both Canadian cohorts, Canadian persons being more dissatisfied with their life since COVID-19 (e.g., 36.1% of participants were very satisfied with their life prior to COVID-19 in the cohort Canada 2 vs. 10.0% upon questionnaire completion, *p* < 0.001). The change was not significant in China (*p* = 0.259), therefore indicating no change in the satisfaction with life before and during the pandemic in China (Figure 5).

### 3.3. COVID Testing

Significant differences in terms of COVID-19 testing were observed between the 3 cohorts, with participants from the Canada 2 cohort being the most tested (41.5%, *p* < 0.001) (Table 3). Additionally, a total of 13 participants tested positive for SARS-CoV-2 in Canada (Canada 1: 9 and Canada 2: 4) and no participants tested positive for COVID-19 in China (Table 3).

### 3.4. COVID-19 Pandemic Concerns and Impacts on Pregnancy Experience

Concerns related to planned birth and delivery changes during the COVID-19 pandemic are presented in Appendix A. Pregnant persons in Canada felt less supported by their primary prenatal care provider(s) and reported more changes in prenatal care support compared to China where there was little to no change in their care/routine (*p* < 0.001) (Appendix A). Those changes were mostly reduction in frequency of perinatal visits (Canada: 24.4% and Canada 2: 17.5%), the replacement of in-person visits with virtual prenatal visits (Canada 1: 17.0% and Canada 2: 16.0%) and cancellation of hospital tours (Canada 1: 13.8% and Canada 2: 14.1%). Regarding delivery, concerns about the partner not being allowed to attend delivery was a very important concern in Canada (Canada 1: 57.9% and Canada 2: 62.6%) compared to China (5.7%) (*p* < 0.001) (Appendix A). Other concerns about insufficient opportunity to initiate breastfeeding or shorter stay in hospital after delivery were all more reported in both Canadian cohorts whereas in China, participants were not concerned by those issues (*p* < 0.001) (Appendix A). Significant differences were also observed in the concern over COVID-19 exposure and complications during delivery between the two Canadian cohorts and the Chinese cohort (Appendix A).

### 3.5. Impact of COVID-19 on Financial Situation and Daily Life

Regarding the pandemic’s effect on participants’ financial situation, 39.7% of pregnant persons reported no change in China when compared to 24.1% in the Canada 1 cohort and 24.0% in the cohort Canada 2 (*p* < 0.001) (Appendix A). Remote work was mostly observed in Canada (*p* < 0.001) but other issues such as job loss or decreased job security did not differ across cohorts (Appendix A). More Chinese participants reported a decreased take-home pay (16.4%) compared to the Canada 1 cohort (12.0%) (*p* < 0.001) and the Canada 2 cohort (8.7%) (*p* = 0.005). Daily routine was more impacted in Canada. Indeed, 46.1% (Canada 1) and 49.6% (Canada 2) of participants reported a severe impact of COVID-19 on their routine compared to 1.8% of Chinese participants (*p* < 0.001) (Appendix A). The impact of the COVID-19 pandemic on medical health care access (not including mental health), on mental health treatment access and on access to family, extended family and social supports was much more detrimental to pregnant persons in Canada compared to China (Appendix A).

### 3.6. Predictors of Maternal Depression

Maternal anxiety (adjusted OR 1.32, 95%CI 1.27–1.38) and stress level (adjusted OR 1.64, 95%CI 1.53–1.77) were significant independent predictors of moderate to severe maternal depression during pregnancy (Table 4). Additionally, when adjusting for all other predictors of depression, Chinese pregnant women were significantly more likely to have moderate to severe depression compared to Canadian pregnant women (cohort 1) (adjusted OR 3.20, 95%CI 1.77–5.78) (Table 4).

## 4. Discussion

To the best of our knowledge, this is the first study directly comparing the impact of the COVID-19 pandemic on maternal mental health, using validated instruments such as EDPS and GAD-7, during pregnancy between Canada and China, two countries that handled the pandemic differently.

Our findings highlight the significant differences existing in levels of depression and anxiety among pregnant persons, which depend on the country of residence and period of recruitment. Indeed, depression and anxiety scores were the highest in the Canada 2 cohort. The mean EDPS score was 8.1 (SD, 5.2) in the Canada 1 cohort, 10.5 (SD, 5.9) in the Canada 2 cohort, and 7.7 (SD, 4.9) in the Chinese cohort (*p*-value Canada 2/China: *p* = 0.005) (Figure 2a). In terms of anxiety, the GAD-7 score was 2.6 (SD, 2.9) in China, 4.3 (SD, 3.9) in the Canada 1 cohort (*p* < 0.001, compared to China) and 5.8 (SD, 5.2) in the Canada 2 cohort (*p* < 0.001, compared to China) (Figure 3a). Satisfaction with life prior to the pandemic compared to when participants completed their questionnaire differed significantly in both Canadian cohorts (e.g., 46.4% of participants were very satisfied with their life prior to COVID-19 in the cohort Canada 1 vs. 23.1% upon questionnaire completion, *p* < 0.001), showing the lasting effects of the COVID-19 pandemic and all related public health measures on quality of life. In contrast, in China, satisfaction with life was not impacted by COVID-19 (*p* = 0.295) (Figure 5). This difference suggests a lower impact of the pandemic in China, which can also be observed in the overall mean level of stress (2.5; SD, 2.1) and concerns relating to pregnancy experience and delivery. The CONCEPTION study herein pointed out several differences in terms of the impact of COVID-19 between the two countries. The prevalence of COVID-19 in our study was 0.6%, with no Chinese participants reporting a COVID-19 positive test. Given the low prevalence of COVID-19 in China, especially at that time, this result is consistent with the literature [29]. The prenatal care system was highly impacted by COVID-19 in Canada. Overall, 19.7% (Canada 1) and 23.3% (Canada 2) of the participants reported no change in prenatal care due to COVID-19, while at the same time 93.2% of Chinese participants reported no change in prenatal care. These results indicate that the presence of COVID-19 and the restrictions implemented to contain the pandemic generated a significant level of distress in pregnant persons, more so in Canada than in China.

When participants from the cohort Canada 2 were recruited, Canada was under severe COVID-19 restrictions (i.e., extended lockdown, curfew, no gatherings between family units), which could explain the observed high level of maternal depression and anxiety among those participants (EDPS, 8.1 (SD, 5.2) and GAD 7, 5.8 (SD, 5.2)). Indeed, when we analyzed the first three waves of the COVID-19 pandemic within the CONCEPTION Study, we found that depressive scores were higher when restrictions were the most severe [18]. At the same time, when Chinese participants were recruited, there were no specific COVID-19 measures in Zhengzhou. This could in part explain the lower levels of anxiety (GAD-7, 2.6 (SD, 2.9)) and stress 2.51 (SD, 2.05). Finally, when pregnant persons from the Canada 1 cohort were recruited, Canada was reopening after the first lockdown, with very little COVID-19 measures and as such had a relatively similar experience as participants in China. Although significant differences were observed between Canada 1 and China in terms of perinatal and delivery outcomes, anxiety and stress were more important in Canada, whereas depression was comparable. Indeed, despite seeing higher anxiety and stress among Canadian pregnant persons, we observed than when adjusting for those variables, Chinese pregnant persons had higher chances of being depressed (adjusted OR 3.20, 95%CI 1.77–5.78) (Table 4). Indeed, our results suggest an increased level of depressive symptoms when strict COVID-19 measures are in place, but as soon as those measures are eased (i.e., end of lockdowns or curfews, gatherings permitted), the level of depressive symptoms decreases. As such, strict lockdowns observed in both Canada and China seem to have a time-dependent effect on maternal mental health. With the decreasing number of confirmed COVID-19 cases and related daily deaths during the recruitment of our Canada 1 and Chinese cohort, it is therefore reasonable to speculate that the prevalence of mental disturbances among pregnant individuals is decreasing as well [29,68]. It is also important to note that the Chinese government acted promptly when faced with the threat of COVID-19, while the provincial and national governments of Canada were slower to impose measures, and public health messaging was often contradictory [2]. The swift course of action observed in China, thought it was perceived as strict, may have led to the improvement of overall mental health throughout the pandemic [2]. Indeed, lower scores of depression and anxiety were reported in China as opposed to Canada [25].

Moreover, when looking at maternal mental health during the pandemic, it is of importance to evaluate the effects of COVID-19 on daily life and income. The impact of the pandemic differed significantly between Canada and China in terms of work situation, food access, daily routine, medical health care access as well as social support, where Canadians were the most affected. As observed in war context, the forced modification of daily life and access to maternal health care can greatly impact maternal mental health during pregnancy [11,12]. Looking at the COVID-19 pandemic with important stressors for pregnant individuals, those restrictions that are imposed on everyday life and access to health care are a key factor to consider when comparing depression, anxiety, and stress among pregnant persons between Canada and China. Due to the mental health burden of COVID-19 restrictions, decision makers, in coordination with health care professionals, should offer mental health assistance to all pregnant individuals. Targeted interventions with already existing e-mental health approaches could help them to cope with the psychological burden of social isolation and governmental restrictions [69,70]. Furthermore, as we move forward and possibly face new waves of the pandemic or other pandemics, healthcare professionals and decision makers should attempt to guarantee unchanged access to prenatal care and delivery conditions for pregnant individuals, as concerns regarding these issues are high and can be a potential source of stress when severe restrictions are in place.

We reported significant differences in the prevalence of medication taken during pregnancy between Canada and China. Only 11.4% of Chinese pregnant persons took at least one prescribed medication, while 57.7% and 55.1% of pregnant persons in the cohorts Canada 1 and 2 did (Appendix A). This difference can be attributed to disparities in tradition between countries. Indeed, Chinese persons are less inclined to be medicated, as per their tradition. OTC medication use was also more common in Canada during pregnancy, at levels of 67.1% (Canada 1) and 60.2% (Canada 2) compared to 5.4% in China (Appendix A). Some cultural differences are also to be noticed with traditional Chinese pregnancy restrictions on behavior and dietary that could have an impact on quality of life compared to Canada [51].

Another marked difference was observed in living arrangements between the two countries. Indeed, to cope with uncomfortable situations such as pregnancy, some pregnant persons choose to go back home to live with their parents/family for support, as shown in Table 1 (living situation, 5.0% (23/484) in China compared to 0.9% (16/1804) in the Canada 1 cohort and 1.5% (2/135) in the Canada 2 cohort). Living with parents and have access to social support is important during pregnancy and was particularly important during the pandemic when social distancing and isolation measures were in place. Social support can buffer the effects of prenatal stress [71,72] and has been shown to mitigate the impacts of prenatal anxiety and depression symptoms on maternal and infant stress response system [73,74]. Decreased prenatal and postnatal anxiety and depression was observed among individuals with higher levels of social support [75,76]. Social support is an important determinant of physical and psychological well-being, especially during pregnancy when individuals take on new responsibilities and roles [77]. Supportive social relationships directly affect mental health by encouraging positive health behaviors, increasing positive feelings, and enhancing emotion regulation [72], and indirectly by reducing the physiological stress response [78].

Canadian and Chinese participants reported depressive scores that are nearly double those of other crisis and non-pandemic periods. In comparison, during the 1998 ice storm crisis in Canada, Laplante et al. reported an EPDS mean score of 5.5 (SD, 2.6) [13]. For non-pandemic periods, a Norwegian study described EDPS scores of 4.8 (SD, 4.3) [79]. Though significant differences exist with Canadian cohorts, the mean depressive score was also high in China (EDPS 7.7, SD, 4.9). This score of depressive symptoms is consistent with that found by other Chinese studies performed during the COVID-19 outbreak (EDPS, 6.4 (SD 4.0); EDPS, 7.7 (SD, 4.4)) [16,17,19]. However, the prevalence of severe depression (EDPS ≥ 13) in our Chinese cohort (17.7%) is consistent with pre-pandemic scores seen in China (16.3% of participants with severe perinatal depression symptoms [80]) but also similar to the remission phase of COVID-19 in China (19.2% of pregnant persons reported depression [29,68]). Looking at anxiety, the prevalence observed in our study seems to be lower than what Lebel et al. reported in Alberta during the first wave (57%) [5]. In the first remission phase after the first wave, the prevalence of moderate to severe anxiety was 8.9% in the Canada 1 cohort and 0.9% in our Chinese group. It was lower than the prevalence of anxiety in other post-pandemic COVID-19 studies in China (9.8%) [29]. This might be due to a decreased fear of the virus as COVID-19 measures had been removed from several month when we started to recruit in China and the looming hope of an upcoming vaccine.

Our study has many strengths. It is the first to assess and compare maternal mental health during the pandemic and therefore compare the impact of the COVID-19 pandemic between Canada and China in a significant sample size. This broad recruitment gave us the opportunity to compare maternal mental health using standardized and validated instruments. In China, pregnant persons were recruited on site by physicians during a visit for heart monitoring in their third trimester, thus explaining this difference in gestational age and trimester between cohorts. All data were collected online, therefore increasing the speed at which the study was performed and allowing us to access real-time results in an ever-evolving pandemic.

Some limits have also been identified. First, despite the large sample size of our Chinese cohort, all participants came from the same region. Although COVID-19 policy was the same in all of China, different levels of restrictions across cities were seen due to differences in the number of COVID-19 cases. Thus, the prevalence of depression and anxiety among pregnant persons may vary across regions of China. It is likely than the length of our questionnaire (≈20 min to complete) has impacted the participation rate. However, it has allowed us to collect many variables of interest that will also be used in the future of the CONCEPTION study to follow pregnant persons and their children over time. We noticed a good completion rate of the questionnaire (85%), even if the impact of its length on participation is difficult to assess. Additionally, the absence of a denominator given the online recruitment in Canada does not allow us to identify differences between participants and non-participants. We acknowledge that our Canadian participants have higher than average household incomes compared to the Canadian population of the same age. Indeed, in 2019, families with children had a median household income of 98,690 CAD, whereas our median salary bracket was 12,000–150,000 CAD [81]. Nonetheless, efforts were made by our recruiting team to provide access to the study to pregnant persons across social media groups as well as through community clinics (lower socioeconomic status) such as the Montreal Diet Dispensary. Furthermore, pregnant participants who decided to complete the questionnaire could be more concerned and worried about the impact of this pandemic on their prenatal/postnatal experience than the general population. Finally, cultural differences between Canada and China may limit the interpretation of data but are of importance to make comparisons and understand the observed results. However, our standardized methodology allowed us to make inter-country comparisons, which is a unique feature of this CONCEPTION study.

As we continue to recruit participants in the CONCEPTION study, we will continue to follow mothers and their children longitudinally. Indeed, we are currently following up children that are 24 months old, in person, to perform neurodevelopmental assessments and study the impact of maternal depression during pregnancy on the child’s development.

## 5. Conclusions

In this first study assessing the impact of COVID-19 on maternal mental health during gestation between Canada and China, we have demonstrated that lockdowns and reopening periods have important effects on levels of depression and anxiety among pregnant persons. However, in reopening phases, mean scores of depression and anxiety remained much higher compared to non-pandemic periods. The burden of COVID-19 containment measures on daily life, social support, prenatal care access and delivery conditions may increase the psychological distress of pregnant individuals. Knowing the potential impacts of such maternal distress during pregnancy on the future neuronal development of their children, there is an urgent need to develop and give access to support for maternal mental health in hopes of reducing the burden of mental health problems during pregnancy.

## Figures and Tables

**Figure 1 ijerph-19-12386-f001:**
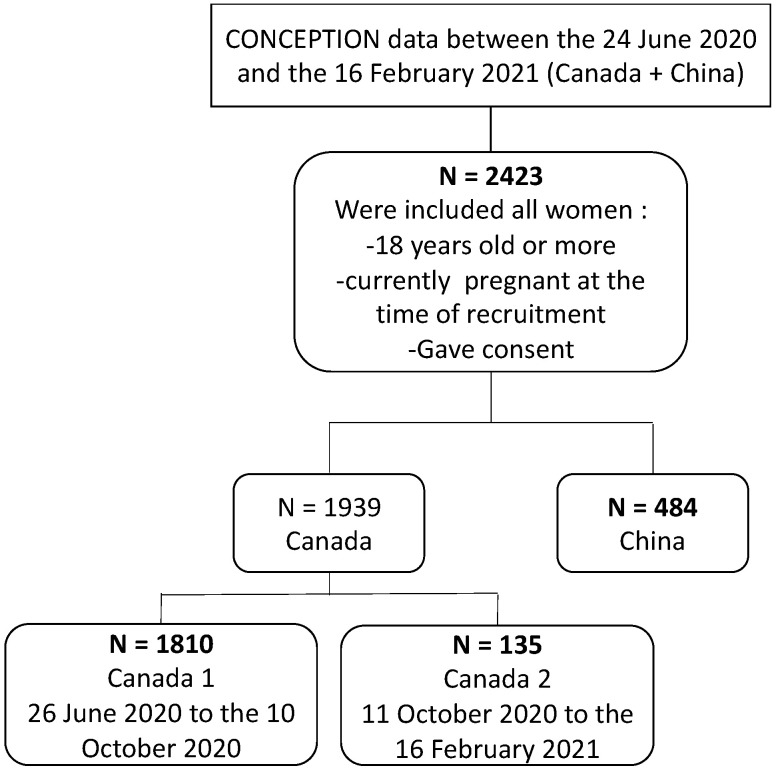
Flow chart of CONCEPTION included pregnant participants.

**Figure 2 ijerph-19-12386-f002:**
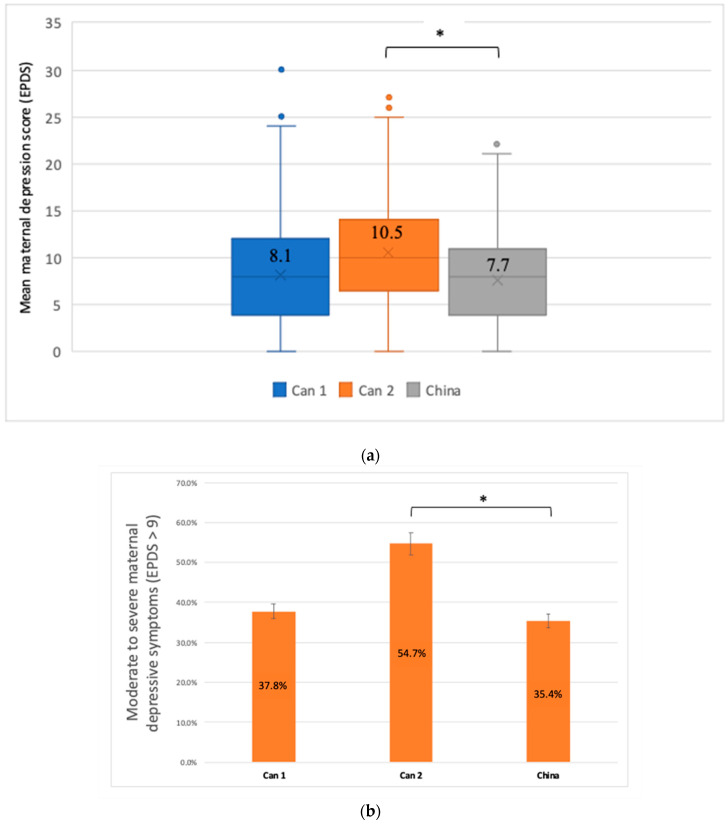
(**a**) Maternal depression using the Edinburgh Postnatal Depression Scale (EPDS). *p* = 0.16 (between Canada 1/China)/*p* = 0.005 (between Canada 2/China). Canada 1 (Can1), *n* = 1592; Canada 2 (Can2), *n* = 117; China, *n* = 413; missing values: Canada 1, *n* = 212; Canada 2, *n* = 18; China, *n* = 71. * *p* < 0.05. Can 1: Canada 1/Can 2: Canada 2. (**b**) Moderate to severe maternal depressive symptoms using the Edinburgh postnatal depression scale (EPDS) cut-off (>9). *p* = 0.92 (between Canada 1/China)/*p* < 0.001 (between Canada 2/China); * *p* < 0.05. Can 1: Canada 1/Can 2: Canada 2. (**c**) Severe maternal depressive symptoms using the Edinburgh postnatal depression scale (EPDS) cut-off (≥13). *p* = 0.08 (between Canada 1/China)/*p* < 0.001 (between Canada 2/China); * *p* < 0.05. Can 1: Canada 1/Can 2: Canada 2.

**Figure 3 ijerph-19-12386-f003:**
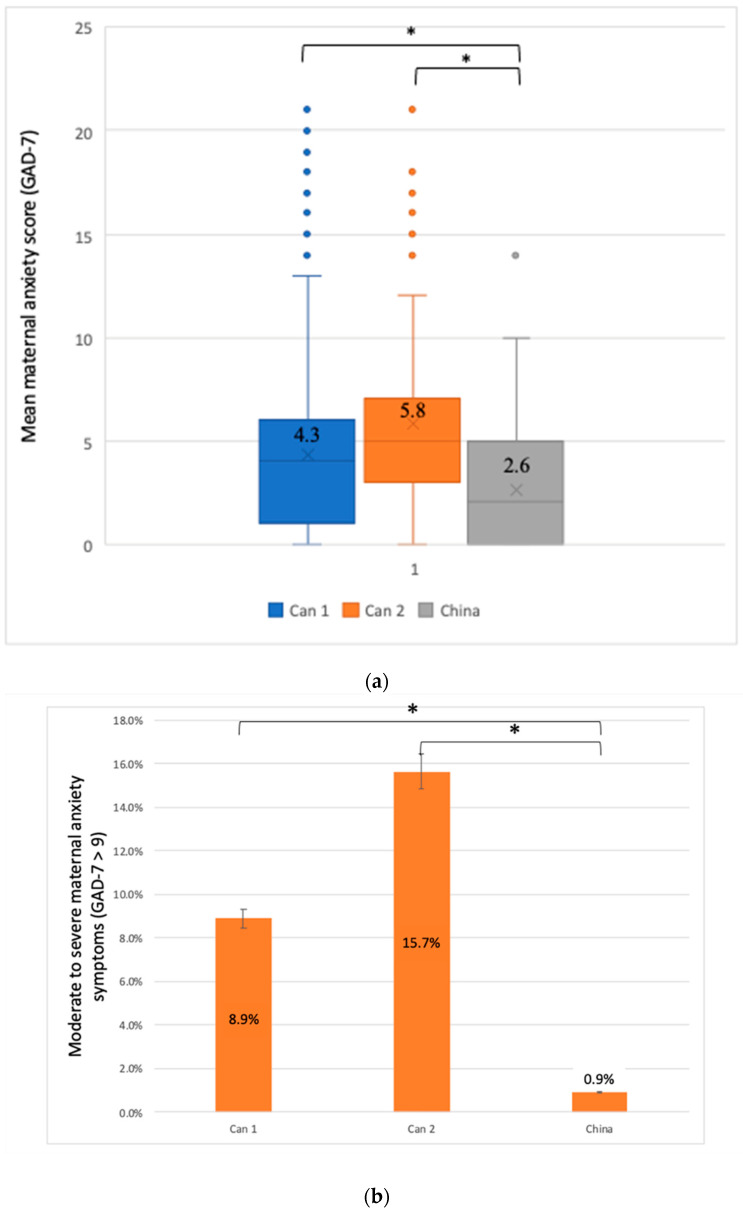
(**a**) Mean maternal anxiety score using the Generalized anxiety disorder 7-item scale (GAD-7) * *p* < 0.001 between Canada 1/China and between Canada 2/China Canada 1 (Can1), *n* = 1563; Canada 2 (Can2), *n* = 115; China, *n* = 438. missing values: Canada 1, *n* = 241; Canada 2, *n* = 20; China, *n* = 46. * *p* < 0.05. Can 1: Canada 1/Can 2: Canada 2. (**b**) Moderate to severe maternal anxiety symptoms using the Generalized Anxiety Disorder 7-item scale (GAD-7) cut-off (>9). * *p* < 0.001 between Canada 1/China and between Canada 2/China. Can 1: Canada 1/Can 2: Canada 2. (**c**) Severe maternal anxiety symptoms using the Generalized Anxiety Disorder 7-item scale (GAD-7) cut-off (>15). (*n* = 0 for China). Can 1: Canada 1/Can 2: Canada 2.

**Figure 4 ijerph-19-12386-f004:**
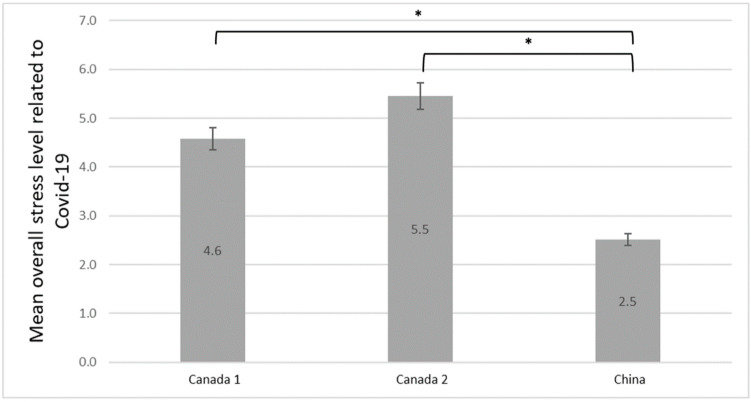
Overall maternal stress level related to COVID-19. * *p* < 0.001 (between Canada 1/China) and between Canada 2/China. Canada 1 (Can1), *n* = 1541; Canada 2 (Can2), *n* = 119; China, *n* = 448. missing values: Canada 1, *n* = 263; Canada 2, *n* = 16; China, *n* = 36.

**Figure 5 ijerph-19-12386-f005:**
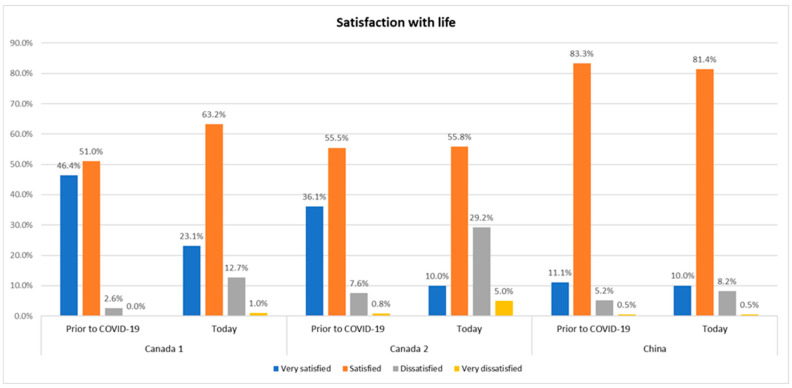
Overall satisfaction with life in pregnant persons. Comparison between prior to vs. since the start of the COVID-19 pandemic stratified by cohorts. Canada 1, *n* = 1646; Canada 2 (Can2), *n* = 120; China, *n* = 429 (missing values: Canada 1, *n* = 158; Canada 2, *n* = 15; China, *n* = 55). Wilcoxon signed rank test—comparing the differences between “prior to COVID-19” and “now” in each cohort. *p* < 0.001 (Canada 1); *p* < 0.001 (Canada 2); *p* = 0.295 (China).

**Table 1 ijerph-19-12386-t001:** Characteristics of pregnant participants.

	Total N = 2423	Canada 1 N = 1804	Canada 2 N = 135	China N = 484	*p*-Value ^+^ CA1/Ch	*p*-Value ^+^ CA2/Ch
Age at recruitment (mean, SD), years	31.6 ± 4.4	31.9 ± 4.3	33.6 ± 4.2	30.2 ± 4.4	**<0.001**	**<0.001**
Missing	21	7	1	13		
Gestational age at recruitment (mean, SD), weeks	26.2 ± 9.9	24.7 ± 9.8	20.7 ± 9.5	33.3 ± 7.7	**<0.001**	**<0.001**
Missing	11	3	-	8		
Pre-pregnancy body mass index, kg/m^2^–(mean, SD)	24.9 ± 5.6	25.4 ± 5.7	26.0 ± 6.2	22.2 ± 4.0	**<0.001**	**<0.001**
Missing	96	15	0	81		
Trimester of pregnancy at the time of survey completion					**<0.001**	**<0.001**
1st trimester	357 (14.8)	305 (16.9)	36 (26.7)	16 (3.3)		
2nd trimester	776 (32.1)	670 (37.2)	64 (47.4)	42 (8.8)		
3rd trimester	1278 (53.1)	826 (45.9)	35 (25.9)	417 (87.8)		
Missing value	11	3	-	8		
Prenatal care follow-up *						
Family physician	712 (27.2)	658 (33.2)	45 (22.6)	9 (1.9)	**<0.001**	**<0.001**
Obstetrician	1632 (62.4)	1090 (55.0)	80 (52.6)	462 (96)	**<0.001**	**<0.001**
Midwife	253 (9.7)	225 (11.4)	25 (16.4)	3 (0.6)	**<0.001**	**<0.001**
Nurse Practitioner	1 (0.04)	0	1 (0.7)	0	-	-
No follow up	16 (0.6)	8 (0.4)	1 (0.7)	7 (1.5)	-	0.014
Missing	6	1	0	5		
Years of education–(mean, SD)	16.2 ± 4.4	16.8 ± 4.5	15.8 ± 6.10	14.4 ± 3.1	**<0.001**	**<0.001**
Missing	96	61	4	31		
Employment status						
Employed	1797 (78.2)	1430 (82.9)	105 (80.8)	262 (59.1)	**<0.001**	**<0.001**
Self-employed	193 (8.4)	151 (8.8)	12 (9.2)	30 (6.8)	0.18	0.33
Student or Intern	78 (3.4)	62 (3.6)	7 (5.4)	9 (2.0)	0.10	0.04
Unemployed	196 (8.5)	49 (2.8)	5 (3.8)	142 (32.1)	**<0.001**	**<0.001**
On welfare	33 (1.4)	32 (1.9)	1 (0.8)	0	-	-
Prefer not to answer	52	28	2	22		
Missing	83	61	3	19		
Ethnic background						
Aboriginal (North American Indians, Métis or Inuit [Inuk])	13 (0.6)	11 (0.6)	1 (0.8)	1 (0.2)	-	-
Asian	455 (19.6)	29 (1.7)	11 (8.5)	415 (91.8)		
Black	20 (0.9)	15 (0.9)	5 (3.9)	-		
Caucasian/White	1714 (73.8)	1608 (92.3)	104 (80.6)	2 (0.4)		
Hispanic	21 (0.9)	20 (1.1)	1 (0.8)	-		
Other	101 (4.3)	11 (0.6)	1 (0.8)	1 (0.2)		
Prefer not to answer	30	29 (1.7)	11 (8.5)	415 (91.8)		
Missing	79	15 (0.9)	5 (3.9)	-		
Living situation					**<0.001**	0.005
Living alone or single mother	39 (1.7)	34 (1.9)	4 (3.1)	1 (0.2)		
Living with a partner/married	2260 (96.3)	1700 (96.9)	124 (94.7)	436 (94.4)		
Living with parents/family	41 (1.7)	16 (0.9)	2 (1.5)	23 (5.0)		
Other	7 (0.3)	4 (0.2)	1 (0.8)	2 (0.4)		
Prefer not to answer	5	4		1		
Missing	71	46	4	21		
Area of residence					**<0.001**	**<0.001**
Rural	313 (13.4)	256 (14.6)	9 (6.9)	48 (10.4)		
Suburban	837 (35.7)	778 (44.5)	52 (40.0)	7 (1.5)		
Urban	1193 (50.9)	716 (40.9)	69 (53.1)	408 (88.1)		
Missing	63	5	21	89		
Household income, CAN$					**<0.001**	**<0.001**
<$30,000	259 (12.4)	39 (2.3)	7 (5.6)	211(71.3)		
$30,000–$60,000	234 (11.2)	170 (10.1)	15 (12.1)	49(16.6)		
$60,001–$90,000	305 (14.5)	275 (16.4)	15 (12.1)	15 (5.1)		
$90,001–$120,000	483 (23.0)	444 (26.5)	31 (25.0)	8 (2.7)		
$120,001–$150,000	333 (15.9)	313 (18.7)	14 (11.3)	6 (2.0)		
$150,000–$180,000	230 (11.0)	207 (12.4)	19 (15.3)	4 (1.4)		
>$180,000	253 (12.1)	227 (13.6)	23 (18.5)	3 (1,0)		
Prefer not to answer	252	81	8	163		
Missing	75	48	3	24		

Numbers are presented as (column percentages) unless stated otherwise. SD, standard-deviation; CAN$, Canadian dollars currency. * Participants could select multiple options. ^+^
*p*-value for comparisons across cohorts. CA1: Canada 1/ CA2: Canada 2/ Ch: China.

**Table 2 ijerph-19-12386-t002:** Pregnancy data.

	Total N = 2423	Canada 1 N = 1804	Canada 2 N = 135	China N = 484	*p*-Value ^+^ CA1/Ch	*p*-Value ^+^ CA2/Ch
First pregnancy						
Yes	1069 (45.6)	772 (44.0)	60 (45.8)	237 (51.6)	0.003	0.24
No	1276 (54.4)	983 (56.0)	71 (54.2)	222 (48.4)		
Missing	78	49	4	25		
Number of children to be born						
Singleton (1 baby)	2304 (98.5)	1723 (98.6)	126 (97.7)	455 (98.7)	-	-
Twins (2 babies)	30 (1.3)	21 (1.2)	3 (2.3)	6 (1.3)		
Multiple (more than 3 babies)	4 (0.2)	4 (0.2)	-	-		
Missing	85	56	6	23		
Current number of children						
0	1185 (50.9)	903 (51.5)	73 (55.7)	209 (47.0)	0.001	0.10
1	844 (36.2)	608 (34.7)	43 (32.8)	193 (43.3)		
≥2	300 (12.9)	242 (13.8)	15 (11.5)	43 (9.7)		
Missing	94	51	4	39		

Numbers are presented as (column percentages) unless stated otherwise. ^+^
*p*-value for comparisons across cohorts. CA1: Canada 1/ CA2: Canada 2/ Ch: China.

**Table 3 ijerph-19-12386-t003:** SARS-CoV-2 testing and COVID-19 diagnosis.

COVID-19 Test	Canada 1 N = 1804	Canada 2 N = 135	China N = 484	Total N = 2423	*p*-Value ^+^ CA1/Ch	*p*-Value ^+^ CA2/Ch
No	1629 (90.4)	79 (58.5)	438 (91.8)	2146 (88.9)	0.34	<0.001
Yes	173 (9.6)	56 (41.5)	39 (8.2)	268 (11.1)		
Positive (if tested)	9 (5.1)	4 (7.1)	0	13 (4.8)	-	-
Missing	2	0	7	9		
Number of immediate family members diagnosed with COVID-19 *						
None	1724 (96.8)	123 (91.8)	449 (99.8)	2296 (97.1)		
1–5	56 (3.1)	11 (8.2)	1 (0.2)	68 (2.9)		
6 or more	1 (0.1)	0	0	1 (0.04)		
No answer	21	1	17	39		
Number of extended family members and/or close friends diagnosed with COVID-19 *						
None	1382 (77.6)	83 (62.4)	454 (99.6)	1919 (81.0)		
1–5	388 (21.8)	48 (36.1)	2 (0.4)	438 (18.5)		
6 or more	11 (0.6)	2 (1.5)	0	13 (0.5)		
No answer	21	2	21	44		

Numbers are presented as (column percentages) unless stated otherwise. * Among participants with answers on the COVID-19 test. ^+^
*p*-value for comparisons across cohorts. CA1: Canada 1/ CA2: Canada 2/ Ch: China.

**Table 4 ijerph-19-12386-t004:** Predictors of maternal depression as defined by the Edinburgh Postnatal Depression Scale.

	Crude Odds Ratio (95%CI)	Adjusted * Odds Ratio (95%CI)
Cohort		
Cohort Canada 1	1.00	1.00
Cohort Canada 2	**1.99 (1.36–2.91)**	1.31 (0.76–2.27)
Cohort China	0.90 (0.72–1.13)	**3.20 (1.77–5.78)**
Anxiety, GAD-7 score **	**1.37 (1.32–1.42)**	**1.32 (1.27–1.38)**
Stress, scale (1–10) **	**1.67 (1.58–1.76)**	**1.64 (1.53–1.77)**
Maternal age, years **	1.01 (0.99–1.03)	1.01 (0.98–1.04)
Body Mass Index, kg.m^2^ **	**1.03 (1.01–1.05)**	1.01 (0.99–1.02)
Weeks’ gestation, weeks **	**1.00 (1.00–1.01)**	1.01 (0.99–1.02)
Employment status		
Employed	1.00	1.00
On welfare or unemployed	1.08 (0.81–1.46)	0.90 (0.56–1.45)
Household income, CAN$		
<$30,000	1.00	1.00
$30,001$60,000	**1.62 (1.10–2.37)**	1.52 (0.82–2.83)
$60,001-$90,000	1.11 (0.77–1.60)	1.04 (0.54–2.03)
$90,001-$120,000	1.70 (0.84–1.63)	1.09 (0.57–2.09)
$120,001-$150,000	1.06 (0.74–1.51)	0.99 (0.50–1.94)
>$150,000	0.76 (0.55–1.07)	0.71 (0.37–1.37)

Legend, Canada 1: participants included between the 26/06/20 and 10/10/20. Canada 2: participants included between the 11/10/20 to 16/02/21. GAD-7: Generalized Anxiety Disorder 7-item scale. CAN$: Canadian dollars. * Adjusted for all variables in the table. ** Continuous variables.

## Data Availability

Anonymized individual-level data from the study including data dictionaries, data collection tools will be made available upon request. Requests for access will be reviewed by a data access committee.

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
