# Peer review of "The Impact of COVID-19 on Maternal Mental Health during Pregnancy: A Comparison between Canada and China within the CONCEPTION Cohort"

_ijerph, 2022, doi:10.3390/ijerph191912386_

Round 1
Reviewer 1 Report
With the collection of web-based data, the manuscript evaluated and compare the impact of COVID-19 public health policies on maternal mental health between Canada and China. While the crises and stressful events during pregnancy and their impacts on the mental health of pregnant is an important topic in public health field research, there is no broad conclusion and implication to explain how the findings contribute to the existing body of knowledge and how to support future studies. However, there are still the following problems in this paper, which need to be further revised by the authors:
Abstract:
The abstract should be revised to explain the knowledge gap and/ or research problem rather than the research process in the current format. Moreover, the study contributes to existing knowledge that must be revealed in the abstract.
Introduction:
This section is too short. The authors missed clearly addressing a question or a problem that has not been answered by any of the existing studies or research within the research field in China and Canada or /and beyond China and Canada. Furthermore, the authors need to justify the similarities between China and Canada in terms of the government structure and policies, people's attitudes and behaviors, and public health strategy regarding pregnancy. So, in the introduction, the authors must clearly state the need for this study (with a range of citations) and what has been solved in the prior studies, and what knowledge gap remains, nonetheless.
Literature review
Generally, the review of the body of literature is still incomplete, and the manuscript has missed reviewing the theoretical background. I see there is no discussion regarding the relevant theories in this field of knowledge.
Methodology:
The research method description is too general. The authors have missed justifying their reasons to employ the current method, and the method's suitability has not been discussed in the study.
Furthermore, the authors employed the questionnaire to collect data, but there is no information on how the questionnaire parts (A, B, C, D) have been designed or developed. Why four parts have been included in the questionnaire? A clear explanation is needed for the questionnaire design or development. Furthermore, the questions must be supported by suitable references and evidence if it has been developed, or the questionnaire must be validated if it has been designed by the authors.
Result
The analysis section needs to be rewritten. The employed analysis method is too general and elementary, while this part is an essential component of manuscripts in high-ranked journals, like IJERPH.
However, the collected data is not convincingly analyzed. The authors need to indicate the conceptual model for the research based on the selected theory(theories) and then introduce the research variables.
However, I guess that the Structural Equation Modeling (SEM) is more appropriate for this study to test and evaluate multivariate causal relationships.
Conclusion
The current discussion is not comprehensive, and it is not convincing enough to implicate the research findings. Thus, it's not evident if this paper offers a significant advancement to the existing literature. Furthermore, there is no information on the suggestions for future studies.
Therefore, the author(s) need to provide comprehensive theoretical and practical implications, and some suggestions for future studies in the discussion part, which are lacking in the study.
Author Response
Reviewer 1:
With the collection of web-based data, the manuscript evaluated and compare the impact of COVID-19 public health policies on maternal mental health between Canada and China. While the crises and stressful events during pregnancy and their impacts on the mental health of pregnant is an important topic in public health field research, there is no broad conclusion and implication to explain how the findings contribute to the existing body of knowledge and how to support future studies. However, there are still the following problems in this paper, which need to be further revised by the authors:
Abstract:
The abstract should be revised to explain the knowledge gap and/ or research problem rather than the research process in the current format. Moreover, the study contributes to existing knowledge that must be revealed in the abstract.
Response: Thank you for your comment and suggestion. As we are limited by the number of words (200 words) we had compacted the abstract to reflect our complex methodology, mainly as was done in our previous study published in the same journal (Bérard et al, Int. J. Environ. Res. Public Health 2022, 19(5), 2926). We understand the importance of explaining the knowledge gap and contribution to existing knowledge, and as such, we have revised the abstract which now reads as follows on page 3 lines 68 to 87:
“The effect of the COVID-19 pandemic on maternal mental health is described in Canada and China but no study has compared the two countries using the same standardized and validated instruments. We aimed to evaluate and compare the impact of COVID-19 public health policies on maternal mental health between Canada and China as we hypothesize that geographical factors and different COVID-19 policies are likely to influence maternal mental health. Pregnant persons >18 years old were recruited in Canada and China using a web-based strategy. All participants recruited between 26/06/20-16/02/21 were analyzed. Self-reported data included sociodemographic variables, COVID-19 experience, and maternal mental health assessments (Edinburgh Perinatal Depression Scale [EPDS], Generalized Anxiety Disorders [GAD-7] scale, stress and satisfaction with life). Analyses were stratified by recruitment cohort, namely: Canada 1 (26/06/20-10/10/20), Canada 2 and China (11/10/20-16/02/21). Overall, 2,423 participants were recruited, respectively 1,804 participants within Canada 1, 135 within Canada 2 and 484 in China. The mean EDPS scores were 8.1 (SD, 5.1) in Canada 1, 8.1 (SD, 5.2) in Canada 2 and 7.7 (SD, 4.9) in China (p-value Canada 2/China: p=0.005). The mean GAD-7 scores were 2.6 (SD, 2.9) in China, 4.3 (SD, 3.8) in Canada 1 (p<0.001) and 5.8 (SD, 5.2) in Canada 2 (p<0.001). When adjusting for stress and anxiety, being part of the Chinese cohort significantly increased the chances of having maternal depression by over 3-fold (adjusted OR 3.20, 95%CI 1.77-5.78). Canadian and Chinese participants reported depressive scores nearly double that of other crises and non-pandemic periods. Lockdowns and reopening periods have an important impact on levels of depression and anxiety among pregnant persons.”
Introduction:
This section is too short. The authors missed clearly addressing a question or a problem that has not been answered by any of the existing studies or research within the research field in China and Canada or /and beyond China and Canada. Furthermore, the authors need to justify the similarities between China and Canada in terms of the government structure and policies, people's attitudes and behaviors, and public health strategy regarding pregnancy. So, in the introduction, the authors must clearly state the need for this study (with a range of citations) and what has been solved in the prior studies, and what knowledge gap remains, nonetheless.
Literature review
Generally, the review of the body of literature is still incomplete, and the manuscript has missed reviewing the theoretical background. I see there is no discussion regarding the relevant theories in this field of knowledge.
Response: We thank you for these suggestions as we believe this will improve the introduction/literature review of our manuscript. We have entirely rewritten the introduction and taken into account both comments above to make the introduction as comprehensive as it can be with the current state of the literature on this topic. The introduction on pages 4-6, lines 98-168 now reads as follows:
“For almost three years, the COVID-19 pandemic has greatly impacted individuals worldwide. The world watched China lock down the border of the provinces of Wuhan and Hubei rapidly in order to contain COVID-19 on January 23rd 2020. Shortly after, on March 11th 2020, the World Health Organization declared COVID-19 a pandemic that would soon ravage the entire world [1]. Governments worldwide reacted to the threat of the pandemic in different ways, with public health measures ranging from regional lockdowns and stay-at-home orders, to extended quarantines, to curfews, to restrictions of services and the shutdown of entire industries among others [2,3]. These measures, along with the uncertainty that the pandemic brought on may have intensified emotional distress, especially among pregnant persons[4,5]. Indeed, it has been observed that the crisis has severely affected people’s mental health leading to a prevalence of depression ranging from 8.3% to 48.3%[6–8].
Exposure to crises and stressful events during pregnancy is known to have long-term effects on the mental health of pregnant individuals as well as on the neuronal development of their offsprings[9–12]. In January 1998, an ice storm crisis in the Canadian provinces of Québec and Ontario resulted in power losses for nearly 3 million people for as long as 40 days. Project Ice Strom by Laplante et al[13] demonstrated that prenatal exposure to a stressful natural disaster was associated with lower cognitive and language abilities in 2-year-old children[14,15]. Additionally, maternal stress and anxiety during pregnancy are also known to be associated with some maternal outcomes such as low birth weight and prematurity[16,17].
To investigate the impact of the COVID-19 pandemic on maternal and perinatal outcomes, the CONCEPTION cohort was initiated by Bérard et al.[18]. Pregnant persons were included worldwide but Chinese and Canadian participants were among the most represented. The effect of the pandemic in both countries is well-described, showing an increased risk of depression and anxiety in pregnant persons that may lead to short and long-term impacts on mothers and children[5,19–24]. To our knowledge, the only study comparing the impact of COVID-19 on pregnant persons between China and Canada, among other countries, is a meta-analysis by Ghazanfarpour et al. conducted in October 2020.[25] They pooled a number of studies across countries and though they had highly heterogenous results, they found that the pooled prevalence of anxiety was of 56% in Canada and 0.3-29% in China while the pooled prevalence of depression was of 37% in Canada and 11-29% in China.[25] Despite the lack of result generalizability and heterogeneity of the pooled studies, they concluded that COVID-19 imposed increased pressure on the emotional well-being of expectant mothers due to the fear of infection, infecting those around them, restricted access to healthcare during their pregnancy and overall restriction from their daily activities.[25] Lok et al. aimed to investigate fear and childbirth experience during both pregnancy and postpartum periods in the COVID-19 crisis recruiting Canadian and Chinese participants however the study is still ongoing [26]. In the general population however, Lee et al. performed a meta-analysis where they compared depression outcomes across country, in relation to the response time of each government respectively [2]. Of note, the prevalence of depressive symptoms was of 21.4% across the 33 included countries [2].They found that early on in the pandemic (before December 2020), governments that acted promptly to reduce the spread of COVID-19 improved not only the physical but also the mental health of their population [2].
The prevalence of COVID-19 and associated measures differed between Canada and China as described in Ghazanfarpour et al. Indeed, Chinese people in the province of Henan only went through one strenuous restriction period between January and April 2020 with the main measures implemented being strict lockdowns, cross-provincial channel barriers and no individual gatherings [27,28]. At the same time, Canada went through two periods of severe COVID-19 restrictions (in spring 2020 and fall/winter 2020) with extended lockdown, curfews, and no individual gatherings [29]. However, given that studies were performed rapidly and on small scales early on in the pandemic, heterogeneity does not allow for direct and robust comparisons to be made between countries.[25]
While there are differences in the way that COVID-19 was handled in both China and Canada, it is important to note some similarities. Indeed, both countries have a similar public health strategy regarding pregnancy with some differences in prenatal visits. Both countries recommend several examinations such as blood tests, an oral glucose tolerance test, HIV screening and screening for colonization with group B streptococcus, as well as physical and ultrasound examinations [30–34]. Moreover, recommendations regarding smoking and the use of alcohol during pregnancy are similar in Canada and China [30,33]. However, the Chinese Medical Association recommends 7 to 11 antenatal care hospital visits [34] while in Canada, 2 to 3 antenatal hospital visits are recommended along with the follow-up by a family physician [35]. Both recommend an increased number of visits for pregnant persons at risk [34,35].
Knowing the numerous complications of maternal mental health on the delivery and the neuronal development of their offspring’s [9–11,14,16,17] and considering both countries dealt with the pandemic in different ways, the CONCEPTION study gave us the opportunity to compare maternal mental health in both China and Canada using the same standardized and validated instruments on large sample size. As the COVID-19 pandemic continues to affect the world population and because geographical factors and different COVID-19 policies influence the prevalence of maternal mental health disorders [2], we aimed to evaluate and compare the impact of COVID-19 measures on maternal mental health to inform healthcare professionals and decision makers on the impact of such decisions, specifically among Canadian and Chinese pregnant persons.”
Methodology:
The research method description is too general. The authors have missed justifying their reasons to employ the current method, and the method's suitability has not been discussed in the study.
Furthermore, the authors employed the questionnaire to collect data, but there is no information on how the questionnaire parts (A, B, C, D) have been designed or developed. Why four parts have been included in the questionnaire? A clear explanation is needed for the questionnaire design or development. Furthermore, the questions must be supported by suitable references and evidence if it has been developed, or the questionnaire must be validated if it has been designed by the authors.
Response: Thank you for your comment. We understand that the methods section seemed brief, and this may be because we referred to a previous manuscript where the methods are described in detail. This manuscript was published a few months ago in the International Journal of Environmental Research and Public Health (IJERPH), entitled The COVID-19 pandemic impacted maternal mental health differently depending on pregnancy status and trimester of gestation by Anick Bérard et al. The latter being the first peer reviewed manuscript to describe the CONCEPTION study in detail.
We have attached the questionnaire in the supplemental files with this rebuttal to make the review process easier, and also for the reader’s better understanding of the study instrument we used. The questionnaire itself is not necessarily split into parts, but rather to explain the contents to the reader, we found it more cohesive to regroup the collected information into categories as is done in most observational studies with self-reported questionnaires, namely maternal sociodemographic characteristics and health history (this allows us to define our cohort in general and compare groups when required by analyses); data on COVID-19 (as we aimed to capture the positivity rate and familial impact of COVID-19 throughout the pandemic); pregnancy experience and changes related to the pandemic as well as the impact of the pandemic on the life of participants (as our experienced team of collaborators developed a tool to measure hardship lived by the pregnant person in relation with the pandemic & public health measures – however this tool is not used in this current study and therefore not described); and lastly, mental health measures using validated tools (to measure the impact of COVID-19 on maternal mental health namely depression, anxiety, stress, and satisfaction with life).
Our questionnaire includes standardized, validated and reliable instruments to compose it, namely the Edinburgh Perinatal Depression Scale, the generalized anxiety disorders scale, and the 4-category ordinal scale to measure satisfaction with life. All measures have been referenced in the manuscript. Our questionnaire was also translated into French, Spanish, Portuguese and Mandarin and then back translated to English to ensure reliability.
We understand that these details are important and as such have added the above-mentioned information, as well as all other details in the Bérard et al. article published in IJERPH that were not mentioned in the present manuscript. As such, we have now modified the “Study design” and “Data collection” sections of the methods, which now read as follows, on pages 7-10, lines 171-247:
“Study design
The CONCEPTION study has been described in detail in Bérard et al.[15]. The recruitment into the CONCEPTION cohort started on 26/06/2020 and is ongoing. The present analysis includes Chinese and Canadian pregnant persons recruited between 26/06/2020 and 16/02/2021. Specifically, Chinese participants were recruited between 11/10/20 and 16/02/21. During this period in Zhengzhou, where all Chinese participants were recruited in hospital, there was no lockdown nor COVID-19 specific restrictions. In an effort to compare the same calendar times between cohorts, we separated our Canadian participants into two cohorts based on their time of recruitment, namely: Canada 1 – participants were enrolled between 26/06/2020 and 10/10/20, Canada 2 – participants were enrolled between 11/10/20 and 16/02/21. Canada 1 captures the summer of reopening in 2020 following the end of the first wave, therefore comparable in terms of COVID-19 conditions and measures with the Chinese cohort; whereas Canada 2 captures the second, and more restrictive lockdown period in Canada.
In Canada, the recruitment of pregnant persons was web-based, using regular daily posting on different social media platforms (Facebook, Twitter, Instagram, TikTok and LinkedIn). To reach as many pregnant persons as possible, the study was promoted by mother/child and pregnancy support groups (e.g. Facebook - Dr. MILK), established hashtag strategies, influencers on Facebook and Instagram with a substantial following, and through communication specialists affiliated with our team's respective Canadian universities. In China, recruitment was done in person in three central hospitals from the Henan Province by investigators when pregnant persons came to the hospital for their clinical follow-up and were handed an electronic device where they completed the questionnaire online independently.
All pregnant individuals aged 18 years or older and able to read one the following language (French, English, Spanish, Portuguese or Mandarin) were eligible. The study instrument was created in English, translated to all available languages and back translated to English in order to ensure their validity. Individual consent was obtained from pregnant persons and data was collected online using the secure platform SurveyMonkey®. All data were then downloaded on a secure server of our hospital at CHU Ste-Justine, Montreal, Quebec. A complete version of the instrument is available in the online supplemental files
Data collection
The baseline questionnaire was tested on 10 English-speaking and 10 French-speaking pregnant persons in order to ensure that questions were understood the same way in the two main languages of use. This questionnaire took on average 25 minutes to complete.
We collected several variables through our study instrument on SurveyMonkey®. A complete version of the instrument is available in the online supplemental files. All following variables were self-reported by pregnant persons.
- A) We first collected general maternal characteristics and health history (past history and since the start of pregnancy) in order to define our study groups in detail. These variables include: 1) general and socio-demographic information: gestational age (continuous), maternal age (continuous), pre-pregnancy height and weight to calculate the body mass index (continuous), ethnicity (Aboriginal, Asian, Black, Caucasian/white, Hispanic, other), annual household income (categorized as <$30,000, $30,000-$60,000, $60,001-$90,000, $90,001-$120,000, $120,001-$150,000, $150,000-$180,000 and >$180,000), years of education (continuous), living situation (with a partner, parents or family, alone), area of residence (urban, rural, suburban), country of residence (Canada, China); 2) Health behaviors including sports, smoking, alcohol and drug use (yes/no); 3) Comorbidities and medication use, including medications available over the counter (OTC); 4) Work/employment status and changes in status following the onset of the COVID-19 crisis and 5) Present experiences related to the COVID-19 pandemic.
- B) We collected data on COVID-19, to measure the positivity rate and familial impact of COVID-19 throughout the pandemic. These variables include: 1) COVID-19 testing (yes/no) and diagnosis by a positive test (yes/no) 2) Number of immediate or extended family member(s) and/or close friends tested positive for COVID-19.
- C) We assessed the impact of the public health measures on the pregnancy experience and changes in birth plans related to the COVID-19 pandemic by collecting information on: 1) Support by primary prenatal care provider(s) and resources available, 2) Type of prenatal classes/information, 3) Support persons not permitted during delivery, 4) Family and friends not permitted in hospital, 5) Separation with newborns after delivery, 6) Concerns about breastfeeding, and 7) All concerns regarding changes in the birth plan and delivery related to COVID-19 were measured on a 4-category ordinal scale; possible responses were “not concerned at all”, “a little concerned”, “moderately concerned” and “very concerned”.
- D) As a proxy for the hardships pregnant participants endur4ed, we asked about the impact of the COVID-19 pandemic on the: 1) financial situation, 2) family income, 3) daily routine, 4) food access, 5) medical health care access excluding mental health, 6) mental health treatment access, 7) access to family, extended family, and non-family social supports, and 8) work situation. Those variables were measured on a 4-category ordinal scale; possible responses were “no change”, ”mild”, “moderate” and “severe”.
- E) We lastly assessed maternal mental health during the COVID-19 pandemic by measuring: 1) Maternal depression during the pandemic, using the Edinburgh Perinatal Depression Scale (EPDS)[28], 2) Anxiety during the pandemic, using the generalized anxiety disorders scale (GAD-7) [29], 3) Satisfaction with life, comparing the time prior to vs. since the start of the COVID-19 pandemic using a 4-category ordinal scale with responses ranging from “very satisfied” to “very unsatisfied” and 4) Stress due to this COVID-19 pandemic using a visual analog scale ranging from 0 (no stress) to 10 (maximum stress). The instruments used to assess maternal mental health are validated and reliable.[28,29] Depression symptoms were measured as continuous variables and further classified as moderate to severe (if EPDS >9) and severe (if EPDS≥13) [22]. Similar to this, anxiety symptoms were classified as moderate to severe (if GAD-7 >9), and severe (if GAD >15)[23].”
Analyses
The analysis section needs to be rewritten. The employed analysis method is too general and elementary, while this part is an essential component of manuscripts in high-ranked journals, like IJERPH.
However, the collected data is not convincingly analyzed. The authors need to indicate the conceptual model for the research based on the selected theory(theories) and then introduce the research variables.
However, I guess that the Structural Equation Modeling (SEM) is more appropriate for this study to test and evaluate multivariate causal relationships.
Response: We thank you for your comment and acknowledge that results presented in this study are indeed mainly descriptive. However, we believe this information is necessary in the understanding of the social impact of COVID-19 on situational changes that in turn have an impact on maternal mental health in pregnancy. We also believe that the presentation of such data collected with appropriate methods and with validated tools (baseline data published in IJERPH), especially in highly ranked journals like IJERPH, allows other research teams to conduct meta-analyses with higher internal and external validity than the one currently available as we have detailed in the introduction.
With your previous comments, we have added a better description of the literature as detailed above as well as a thorough justification of our study variables. As such, we feel that the analyses have a better context now. With this said, we have specified why we chose to analyse our data the way we did, specifically when it pertains to the cohorts used for comparison. Indeed, the analyses section has now been introduced as follows, in the methods section on page 9, lines 250-255:
“Given that pandemic waves have had an impact on maternal depression in Canada [15] and given that the time of response from the governments when facing the pandemic has an impact on mental health in the general population (3) analyses were stratified according to the recruitment cohort (Canada 1, Canada 2, and China). As described above, each cohort was recruited at a specific time-period during the pandemic and mothers had therefore lived the pandemic differently.”
As for the multivariate analyses, in this study, we aimed to quantify determinants of depression, defined as an EDPS score >9. It is important to note that confounding variables were defined a priori and are based on both the known literature on depression predictors but also on our findings within the CONCEPTION Cohort in Bérard et al (IJERPH, 2022). Given our understanding and theoretical background of depression and its’ predictors, we do not believe that the use of SEM would be appropriate in this context. We have made this specification in the analysis section of the methods, on pages 9-10, lines 265-272:
“We quantified the determinants of depression (EPDS>9) during pregnancy by quantifying crude and multivariate associations using logistic regression models, considering the cohort of recruitment, maternal anxiety (continuous), maternal stress (continuous), maternal age (continuous), pre-pregnancy body mass index (continuous), weeks’ gestation at recruitment (continuous), employment status (employed [reference], unemployed or on welfare), years of education (continuous), and annual household income (categorized as defined above) as predictor variables. These adjustment variables were determined a priori and are based on our previous findings within the CONCEPTION Cohort.[15]”
Discussion
The current discussion is not comprehensive, and it is not convincing enough to implicate the research findings. Thus, it's not evident if this paper offers a significant advancement to the existing literature. Furthermore, there is no information on the suggestions for future studies.
Therefore, the author(s) need to provide comprehensive theoretical and practical implications, and some suggestions for future studies in the discussion part, which are lacking in the study
Response: We thank you for your comment. As we took into consideration your previous comments as well as those from Reviewer 2, we reworked the manuscript accordingly, which we believe has led to the improvement of the discussion section.
The discussion has mostly been modified specifically, on pages 23-25, lines 454 to 536 which now reads as follows:
“To our knowledge, this is the first study directly comparing the impact of the COVID-19 on maternal mental health, using validated instruments such as EDPS and GAD-7, during pregnancy between Canada and China, two countries that handled the pandemic differently.
Our findings highlight the significant differences in levels of depression and anxiety among pregnant persons, which depend on the country of residence and period of recruitment. Indeed, depression and anxiety scores were the highest in the Canada 2 cohort. The mean EDPS score was 8.1 (SD, 5.2) in the Canada 1 cohort, 10.5 (SD, 5.9) in the Canada 2 cohort, and 7.7 (SD, 4.9) in the Chinese cohort (p-value Canada 2/China: p=0.005) (Figure 2). In terms of anxiety, the GAD-7 score was 2.6 (SD, 2.9) in China, 4.3 (SD, 3.9) in the Canada 1 cohort (p<0.001, compared to China), and 5.8 (SD, 5.2) in the Canada 2 cohort (p<0.001, compared to China) (Figure 3). Satisfaction with life prior the pandemic compared to when participants completed their questionnaire differed significantly in both Canadian cohorts (e.g. 46.4% of participants were very satisfied with their life prior to COVID-19 in the cohort Canada 1 vs 23.1% upon questionnaire completion, p<0.001), showing lasting effects of the COVID-19 pandemic and all related public health measures on quality of life. In contract, in China, satisfaction with life was not impacted by COVID-19 (p=0.295) (Figure 5). This difference suggests a lower impact of the pandemic in China, which can also be observed in the overall mean level of stress (2.5; SD, 2.1) and concerns on pregnancy experience and delivery. The CONCEPTION study herein pointed out several differences in terms of the impact of COVID-19 between the two countries. The prevalence of COVID-19 in our study was 0.6% with no Chinese participants reporting a COVID-19 positive test. Given the low prevalence of COVID-19 in China, especially at that time, this result is consistent with the literature [27]. The prenatal care system was very impacted by COVID-19 in Canada. Overall, 19.7% (Canada 1) and 23.3% (Canada 2) of participants reported no change in prenatal care due to COVID-19 while at the same time, 93.2% of Chinese participants reported no change in prenatal care. These results indicate that the presence of COVID-19 and the restrictions associated to contain the pandemic generated an important distress in pregnant persons, more so in Canada than in China.
When participants from the cohort Canada 2 were recruited, Canada was under severe COVID-19 restrictions (i.e. extended lockdown, curfew, no gatherings between family units) which could explain the observed high level of maternal depression and anxiety among those participants (EDPS, 8.1 [SD, 5.2] and GAD 7, 5.8 [SD, 5.2]). Indeed, when we analyzed the first three waves of the COVID-19 pandemic within the CONCEPTION Study, we found that depressive scores were higher when restrictions were the most severe[18]. At the same time, when Chinese participants were recruited, there were no specific COVID-19 measures in Zhengzhou. This could in part explain the lower levels of anxiety (GAD-7, 2.6 [SD, 2.9]) and stress 2.51 (SD, 2.05). Finally, when pregnant persons from the Canada 1 cohort were recruited, Canada was reopening after the first lockdown, with very little COVID-19 measures and as such had a relatively similar experience as participants in China. Although significant differences were observed between Canada 1 and China in terms of perinatal and delivery outcomes, anxiety and stress were more important in Canada, whereas depression was comparable. Indeed, despite seeing higher anxiety and stress among Canadian pregnant persons, we observed than when adjusting for those variables, Chinese pregnant persons had higher chances of being depressed (adjusted OR 3.20, 95%CI 1.77-5.78) (Table 4). Indeed, our results suggest an increased level of depressive symptoms when strict COVID-19 measures are in place, but as soon as those measures are eased (i.e. end of lockdowns or curfews, gatherings permitted), the level of depressive symptoms decreases. As such, strict lockdowns observed in both Canada and China seem to have a time-dependent effect on maternal mental health. With the decreasing number of confirmed COVID-19 cases and related daily deaths during the recruitment of our Canada 1 and Chinese cohort, it is therefore reasonable to speculate that the prevalence of mental disturbances among pregnant individuals is decreasing as well[27,38]. It is also important to note that the Chinese government acted promptly when faced with the threat of COVID-19, while the provincial and national governments of Canada were slower to impose measures, and public health messaging was often contradictory [2]. The swift course of action observed in China, thought it was perceived as strict may have led to the improvement of overall mental health throughout the pandemic. [2] Indeed, lower scores of depression and anxiety were reported in China as opposed to Canada.[25]
Moreover, when looking at maternal mental health during the pandemic, it is of importance to evaluate the effects of COVID-19 on daily life and income. The impact of the pandemic differed significantly between Canada and China in terms of work situation, food access, daily routine, medical health care access as well as social support, where Canadians were the most affected. As observed in war context, the forced modification of daily life and access to maternal health care can greatly impact maternal mental health during pregnancy [11,12]. Looking at this COVID-19 pandemic with important stressors for pregnant individuals, those restrictions that are imposed on everyday life and access to health care are a key factor to consider when comparing depression, anxiety, and stress among pregnant persons between Canada and China. Because of the mental health burden of COVID-19 restrictions, decision makers, in coordination with health care professionals, should offer mental health assistance to all pregnant individuals. Targeted interventions, with already existing e-mental health approaches could help them to cope with the psychological burden of social isolation and governmental restrictions [39,40]. Furthermore, as we move forward and possibly face new waves of the pandemic or other pandemics all together, healthcare professionals and decision makers should attempt to guarantee an unchanged access to prenatal care and delivery conditions for pregnant individuals as concerns regarding those issues are high and can be a potential source of stress when severe restrictions are in place.
We reported significant differences in the prevalence of medication taken during pregnancy between Canada and China. Only 11.4% of Chinese pregnant persons took at least one prescribed medication, while 57.7% and 55.1% of pregnant persons in the cohorts Canada 1 and 2 did (Supp. Table S1). This difference can be attributed to disparities in tradition between countries. Indeed, Chinese persons are less inclined to be medicated as per their tradition. OTC medication use was also more common in Canada during pregnancy: 67.1% (Canada 1) and 60.2% (Canada 2) compared to 5.4% in China (Supp. Table S1). Some cultural differences are also to be noticed with traditional Chinese pregnancy restrictions on behavior and dietary that could have an impact on quality of life compared to Canada [41].”
Lastly, we discussed implications and added future directions to our discussion:
- On pages 24-25 lines 519-527:
“Because of the mental health burden of COVID-19 restrictions, decision makers, in coordination with health care professionals, should offer mental health assistance to all pregnant individuals. Targeted interventions, with already existing e-mental health approaches could help them to cope with the psychological burden of social isolation and governmental restrictions [39,40]. Furthermore, as we move forward and possibly face new waves of the pandemic or other pandemics all together, healthcare professionals and decision makers should attempt to guarantee an unchanged access to prenatal care and delivery conditions for pregnant individuals as concerns regarding those issues are high and can be a potential source of stress when severe restrictions are in place.
- And on page 27, lines 602-605:
“As we continue to recruit participants in the CONCEPTION Study, we will continue to follow mothers and their children longitudinally. Indeed, we are currently following up children that are 24 months old, in person, to perform neurodevelopmental assessments and study the impact of maternal depression during pregnancy on the child’s development.”
Reviewer 2 Report
The manuscript evaluates maternal mental health during the pandemic and therefore compare the impact of the COVID-19 pandemic between Canadian and Chinese persons.
The author has highlighted the significant differences in levels of depression and anxiety among pregnant persons, which depend on the country of residence and period of recruitment.
The author has also provided detailed information in the form of Self-reported data taking into consideration various factors such sociodemographic variables, COVID-19 experience, maternal mental health assessments (Edinburgh Perinatal Depression Scale [EPDS], Generalized Anxiety Disorders [GAD-7] scale, stress and satisfaction with life), pre- natal care/birth plan changes and impact of COVID-19 on their living situation.
Although, author has provided detailed information regarding self reported data but more information needs to be provided regarding COVID-19 policies and how they impacted on maternal mental health.
Also, information needs to be provided to highlight role of healthcare professionals and decision makers on daily life, social support, prenatal care access and delivery conditions of pregnant individuals in Canada and China.
Author Response
Reviewer 2:
The manuscript evaluates maternal mental health during the pandemic and therefore compare the impact of the COVID-19 pandemic between Canadian and Chinese persons.
The author has highlighted the significant differences in levels of depression and anxiety among pregnant persons, which depend on the country of residence and period of recruitment.
The author has also provided detailed information in the form of Self-reported data taking into consideration various factors such sociodemographic variables, COVID-19 experience, maternal mental health assessments (Edinburgh Perinatal Depression Scale [EPDS], Generalized Anxiety Disorders [GAD-7] scale, stress and satisfaction with life), pre- natal care/birth plan changes and impact of COVID-19 on their living situation.
Although, author has provided detailed information regarding self reported data but more information needs to be provided regarding COVID-19 policies and how they impacted on maternal mental health.
Response: Thank you for your comment. COVID-19 policies were indeed quite different from one country to another, specifically between China and Canada, which is the premise of this study. We have added details throughout the manuscript to better describe these differences, in the following sections:
In the Introduction, on page 5 lines 140-148, which now reads as follows:
“The prevalence of COVID-19 and associated measures differed between Canada and China as described in Ghazanfarpour et al. Indeed, Chinese people in the province of Henan only went through one strenuous restriction period between January and April 2020 with the main measures implemented being strict lockdowns, cross-provincial channel barriers and no individual gatherings [27,28]. At the same time, Canada went through two periods of severe COVID-19 restrictions (in spring 2020 and fall/winter 2020) with extended lockdown, curfews, and no individual gatherings [29]. However, given that studies were performed rapidly and on small scales early on in the pandemic, heterogeneity does not allow for direct and robust comparisons to be made between countries.[25]
The impact of such measures on maternal mental health has been precised in the Discussion, page 24 lines 496 to 500:
“Indeed, our results suggest an increased level of depressive symptoms when strict COVID-19 measures are in place, but as soon as those measures are eased (i.e. end of lockdowns or curfews, gatherings permitted), the level of depressive symptoms decreases. As such, strict lockdowns observed in both Canada and China seem to have a time-dependent effect on maternal mental health.”
Also, information needs to be provided to highlight role of healthcare professionals and decision makers on daily life, social support, prenatal care access and delivery conditions of pregnant individuals in Canada and China.
Response: Thank you for this suggestion. We indeed found it important to discuss how healthcare professionals can offer support to pregnant individuals, as such we have added the following section to the Discussion, on page 24, lines 519-527, which now reads as follows:
“Because of the mental health burden of COVID-19 restrictions, decision makers, in coordination with health care professionals, should offer mental health assistance to all pregnant individuals. Targeted interventions, with already existing e-mental health approaches could help them to cope with the psychological burden of social isolation and governmental restrictions [39,40]. Furthermore, as we move forward and possibly face new waves of the pandemic or other pandemics all together, healthcare professionals and decision makers should try to guarantee an unchanged access to prenatal care and delivery conditions for pregnant individuals as concerns regarding those issues are high and can be a potential source of stress when severe restrictions are in place.”
Round 2
Reviewer 1 Report
After reading the revised manuscript, I found three comments were not appropriately addressed by the authors.
Comment: the authors need to justify the similarities between China and Canada in terms of the government structure and policies, people's attitudes and behaviors, and public health strategy regarding pregnancy. So, in the introduction, the authors must clearly state the need for this study (with a range of citations) and what has been solved in the prior studies, and what knowledge gap remains, nonetheless.
The authors highlighted the two countries' recommendations on several examinations (such as blood tests, an oral glucose tolerance test, HIV screening, and screening for colonization with group B streptococcus, as well as physical and ultrasound examinations). Indeed, the examination tests are diagnostic tools, so can not demonstrate the similarities between public health strategies in the two countries. Thus, it is not convincing enough for a selection of China and Canda as case studies
Comment: Generally, the review of the body of literature is still incomplete, and the manuscript has missed reviewing the theoretical background. I see there is no discussion regarding the relevant theories in this field of knowledge.
The authors improved the introduction/literature review of the manuscript, but there is no discussion about the theoretical background, and they employed theory to conduct the study.
Comments: The research method description is too general. The authors have missed justifying their reasons to employ the current method, and the method's suitability has not been discussed in the study.
The author mentioned that the methods section seemed brief, and this may be because we referred to a previous manuscript where the methods are described in detail.
I am still confused with the authors' justification as this part is essential for a high-quality article in Q1 Journal like IJERPH.
Based on them, and on my own reading, I regret that I cannot accept the manuscript as it stands.
Author Response
Reviewer 1
After reading the revised manuscript, I found three comments were not appropriately addressed by the authors.
Comment: the authors need to justify the similarities between China and Canada in terms of the government structure and policies, people's attitudes and behaviors, and public health strategy regarding pregnancy. So, in the introduction, the authors must clearly state the need for this study (with a range of citations) and what has been solved in the prior studies, and what knowledge gap remains, nonetheless.
The authors highlighted the two countries' recommendations on several examinations (such as blood tests, an oral glucose tolerance test, HIV screening, and screening for colonization with group B streptococcus, as well as physical and ultrasound examinations). Indeed, the examination tests are diagnostic tools, so can not demonstrate the similarities between public health strategies in the two countries. Thus, it is not convincing enough for a selection of China and Canada as case studies.
Response: We thank you for the suggestion to provide further context on policies and behaviors as we believe this has improved our introduction. We have done an extensive literature review on public health strategies, both pertaining to COVID-19 and general health in China and Canada, which we will detail below.
We first discussed the differences between China and Canada in terms of their approach to COVID-19. Indeed China opted for the zero COVID strategy, while Canada was more heterogeneous. Indeed, Canadian provinces are in charge of core public health decisions, and as such, some opted the zero COVID approach, while others opted for a mitigation strategy. This section of the introduction, on pages 5-6, lines 146-163 now reads as follows:
“The prevalence of COVID-19 and associated public health measures to contain the pandemic differed between Canada and China as described in Ghazanfarpour et al.[25]. China was the first country affected by COVID-19 in the end of December 2019. On the one hand, the Chinese government rapidly implemented a zero COVID strategy in hopes of eradicating the virus form the country [27,28].To achieve this, they implemented strict lockdowns, cross-provincial channel barriers and imposed no individual gatherings starting in January 2020 [29,30]. Measures were then eased on April 2020, however the zero COVID strategy remained active with a precise system of prevention and controls, including mass testing and isolation of positive cases in quarantine centers[27,28,31]. At the same time, there was restricted access to hospital centers for routine visits or follow-up of chronic illnesses, which includes obstetrical follow-ups [27,31–33]. On the other hand, Canada attempted to mitigate the spread of the virus and associated hospitalizations/deaths while also balancing economic activities. However, the zero COVID-19 approach was not considered in most Canadian provinces (i.e. Alberta, British Columbia, Manitoba, Ontario, Quebec and Saskatchewan), rather only in the territories (i.e. Northwest Territories, Nunavut, and Yukon) as well as the Maritimes (i.e. New Brunswick, Newfoundland and Labrador, and Prince Edward Island)[34]. Through Canadian public health mitigation strategies, most Canadian provinces went through two periods of severe COVID-19 restrictions to limit the spread of the virus (in spring 2020 and fall/winter 2020) with extended lockdowns, curfews, and no individual gatherings [32,34].”
We then discussed the maternal healthcare policies and programs in both countries, which highlighted similarities. Additionally, we discussed how the pregnancy follow-up was affected during the COVID-19 pandemic. This section of the introduction, on page 6, lines 165-189 now reads as follows:
“While there are stark differences in the way that the COVID-19 pandemic was handled in China and Canada, it is important to note some similarities when it comes to more general public health strategies. First, maternal healthcare has been made a priority in both countries. Indeed, in the last two decades, China has made one of its priorities to reduce maternal and under-5 mortality and now has a similar maternal mortality rate (18.3 per 100,000 livebirths in 2018) as Canada (8.4 per 100,000 livebirths in 2020) and the Western world (12.0 per 100,000 livebirths in 2017) [35–38]. China implemented the Five Strategies for Maternal and Newborn Safety and the Healthy China initiative in 2016 [36,39]; while in Canada The Program for Prevention of Maternal Morbidity and Mortality was launched in 2016 [40]. When looking specifically at the medical follow-up during pregnancy both countries recommend several examinations such as blood tests, an oral glucose tolerance test, HIV screening as well as regular physical and ultrasound examinations [41–45]. Moreover, recommendations regarding smoking and the use of alcohol during pregnancy are similar in Canada and China [41,44]. The Chinese Medical Association recommends 7 to 11 antenatal care hospital visits [45] and in Canada, 2 to 3 antenatal hospital visits are recommended along with the regular follow-up by a family physician or obstetrician, which adds up to the same number of follow-ups overall [46]. Both recommend an increased number of visits for pregnant persons at risk and towards the end of pregnancy [45,46]. In terms of access to this antenatal follow-up, Chinese persons have the choice between universal health coverage which occurs in public institutions and are covered by social security contributions, or the private/international institutions which typically entails the subscription to a health insurance, either paying for the totality of healthcare services or a portion of it [35,47,48]. Canadian persons on the other hand have universal access for all their healthcare services, funded through tax-payers, available to them [49]. This said, when it comes to COVID-19, both countries limited onsite hospital visits for pregnant persons and promoted online counseling and training programs during lockdowns [32,33].”
Additionally, we highlighted cultural differences and how this may impact pregnancy differently in China compared to Canada. This section of the introduction, on page 6, lines 190-195 now reads as follows:
“Differences in cultural habits and beliefs during pregnancy can be observed. In the Chinese culture, the pregnant person is considered vulnerable and requires continued rest[50]. As a consequence, Chinese pregnant individuals exist the work force early on in their pregnancy compared to Canadians [50]. Also, traditional taboos in China, such as “not walking too fast” to avoid spontaneous miscarriage and restriction on certain type of food or activities can make pregnancy a different experience in the two countries [50,51].”
To echo these cultural differences and beliefs during pregnancy as well as to put our results into context, this is also highlighted in the discussion section on page 26 lines 687 to 707:
“This difference can be attributed to disparities in tradition between countries. Indeed, Chinese persons are less inclined to be medicated as per their tradition. OTC medication use was also more common in Canada during pregnancy: 67.1% (Canada 1) and 60.2% (Canada 2) compared to 5.4% in China (Supp. Table S1). Some cultural differences are also to be noticed with traditional Chinese pregnancy restrictions on behavior and dietary that could have an impact on quality of life compared to Canada [51].
Another marked difference is observed in living arrangements between the two countries. Indeed, to cope with uncomfortable situations such as pregnancy, some pregnant persons choose to go back home to live with their parents/family for support as shown in Table 1 (living situation, 5.0% [23/484] in China compared to 0.9% [16/1804] in the Canada 1 cohort and 1.5% [2/135] in the Canada 2 cohort). Living with parents and have access to social support is important during pregnancy and was particularly important during the pandemic when social distancing and isolation were in place. Social support can buffer the effects of prenatal stress[71,72], and has been shown to mitigate the impacts of prenatal anxiety and depression symptoms on maternal and infant stress response system[73,74]. Decreased prenatal and postnatal anxiety and depression was observed among individuals with higher levels of social support[75,76]. Social support is an important determinant of physical and psychological well-being, especially during pregnancy when individuals take on new responsibilities and roles[77]. Supportive social relationships directly affect mental health by encouraging positive health behaviors, increasing positive feelings, and enhancing emotion regulation[72], and indirectly by reducing the physiological stress response[78].”
Following the discussion of existing literature, we also made sure to explain the knowledge gap and the importance of conducting this study. This section of the introduction, on page 5, lines 120-145 now reads as follows:
“To investigate the impact of the COVID-19 pandemic on maternal and perinatal outcomes, the CONCEPTION cohort was initiated by Bérard et al.[18]. Pregnant persons were included worldwide but Chinese and Canadian participants were among the most represented. The effect of the pandemic in both countries is well-described individually, showing an increased risk of depression and anxiety in pregnant persons that may lead to short and long-term impacts on mothers and children[5,19–24]. To our knowledge, the only study comparing the impact of COVID-19 on pregnant persons between China and Canada, among other countries, is a meta-analysis by Ghazanfarpour et al. conducted in October 2020.[25] They pooled a number of studies across countries and though they had highly heterogenous results, they found that the pooled prevalence of anxiety was of 56% in Canada and 0.3-29% in China, while the pooled prevalence of depression was of 37% in Canada and 11-29% in China.[25] Despite the lack of result generalizability and heterogeneity of the pooled studies, specifically in relation to the tools used to assess anxiety and depression, they concluded that COVID-19 imposed increased pressure on the emotional well-being of expectant mothers due to the fear of infection, infecting those around them, restricted access to healthcare during their pregnancy and overall restriction from their daily activities.[25] Lok et al. aimed to investigate fear and childbirth experience during both pregnancy and postpartum periods in the COVID-19 crisis recruiting Canadian and Chinese participants, however the study is still ongoing [26]. In the general population however, Lee et al. performed a meta-analysis where they compared depression outcomes across country, in relation to the response time of each government respectively [2]. Of note, the prevalence of depressive symptoms was of 21.4% across the 33 included countries [2].They found that early on in the pandemic (before December 2020), governments that acted promptly to reduce the spread of COVID-19 improved not only the physical but also the mental health of their population [2]. However, given that studies were performed rapidly and on small scales early on in the pandemic, heterogeneity does not allow for direct and robust comparisons to be made between countries.[25] As we will continue to evaluate the impact of public health measures and the pandemic itself on maternal and perinatal outcomes, it is of utmost importance to generate scientific findings using validated tools to measure maternal mental health outcomes specifically.”
Comment: Generally, the review of the body of literature is still incomplete, and the manuscript has missed reviewing the theoretical background. I see there is no discussion regarding the relevant theories in this field of knowledge.
The authors improved the introduction/literature review of the manuscript, but there is no discussion about the theoretical background, and they employed theory to conduct the study.
Response: Thank you for your comment. To build on the previous revisions described above, we believe the introduction now highlights: 1) the current state of the literature as well as knowledge gaps, 2) how differently both countries handled the pandemic, and 3) the similarities in terms of public health policies pertaining to maternal mental health and obstetrical follow-up. We have now provided the theoretical background of the study for the reader, based on the literature review. This section of the introduction, on page 7, lines 197-219 now reads as follow:
“The COVID-19 pandemic generated uncertainty since January 2020, which has been directly correlated with increased stress in the general and pregnant populations [5–7,9,19–21,24,25]. This stress is the result of limited accurate information on the virus, limited availability of proven therapeutics, contradicting vaccination campaigns, and strong public policies to reduce the transmission of the virus [27,28,31,52]. For example, Lebel et al. surveyed pregnant individuals at the beginning of the pandemic in Canada and reported elevated clinically relevant symptoms of depression (37%) and anxiety (57%) comparted to similar pre-pandemic scores [5]. Moreover, the implementation of COVID-19 measures in the long run could increase the prevalence of depressive and anxiety symptoms because of social isolation and limited access to social and medical support [2,25]. As such, based on the current state of the literature summarized herein, we hypothesized that the more restrictive and lasting the COVID-19 measures, the greater the impact on maternal mental health. Therefore, considering Canada and China continue to deal with the pandemic in different ways but have overall similar maternal health policies and public health strategies regarding pregnancy, the CONCEPTION study gave us the opportunity to answer this knowledge gap on the direct impact, specifically assessed with validated and standardized instruments in both countries. This is currently lacking in the present literature. Indeed, knowing the numerous complications of maternal mental health on the delivery and the neuronal development of their offspring’s [9–11,14,16,17], and because the COVID-19 pandemic continues to affect the world, we aimed to evaluate and compare the impact of COVID-19 measures on maternal mental in order to inform healthcare professionals and decision makers on the impact of such decisions, specifically among Canadian and Chinese pregnant persons.”
Comment: The research method description is too general. The authors have missed justifying their reasons to employ the current method, and the method's suitability has not been discussed in the study.
The author mentioned that the methods section seemed brief, and this may be because we referred to a previous manuscript where the methods are described in detail.
I am still confused with the authors' justification as this part is essential for a high-quality article in Q1 Journal like IJERPH.
Response: We apologize for the confusion, we had originally referenced the first CONCEPTION paper by Bérard et al. (IJERPH, 2022) to keep our first version of the methods brief. However, we have now revised and improved this section in order to better explain the reasons why we conducted the CONCEPTION study as we did, and have provided more background theory on the standardized tools we used.
As we determined that the waves of the pandemic had a differential impact on maternal mental health, we thought it was important to compare cohorts based on calendar time and separate our Canadian participants into two cohorts based on their time of recruitment. This section of the methods, on page 8, lines 224-237 reads as follows:
“The present analysis includes Chinese and Canadian pregnant persons recruited between 26/06/2020 and 16/02/2021. Specifically, Chinese participants were recruited between 11/10/20 and 16/02/21. During this period in Zhengzhou, where all Chinese participants were recruited in university-affiliated hospitals, there was no lockdown nor any COVID-19 specific restrictions. In our previous study within the CONCEPTION cohort, we determined that the waves of the pandemic had a differential impact on maternal mental health [18], as such, it was important to compare cohorts based on calendar time. As such, we separated our Canadian participants into two cohorts based on their time of recruitment, namely: Canada 1 – participants were enrolled between 26/06/2020 and 10/10/20, Canada 2 – participants were enrolled between 11/10/20 and 16/02/21. Canada 1 captures the summer of reopening in 2020 following the end of the first wave, therefore comparable in terms of COVID-19 conditions and measures with the Chinese cohort, whereas Canada 2 captures the second and more restrictive lockdown period in Canada.”
We also discussed the suitability of online recruitment in Canada, the addition of local, in-person recruitment in Canada as well as the difference with the recruitment method in China. This section of the methods, on pages 8 and 9, lines 238 to 258 now reads as follows:
“In Canada, the recruitment of pregnant persons was web-based, using regular daily posting on different social media platforms (Facebook, Twitter, Instagram, TikTok and LinkedIn). To reach as many pregnant persons as possible, the study was promoted by mother/child and pregnancy support groups (e.g. Facebook - Dr. MILK as well as privately run “mom” groups), established hashtag strategies, influencers on Facebook and Instagram with a substantial following, and through communication specialists affiliated with our team's respective Canadian universities. Online recruitment using anonymized surveys has been used in several other similar studies [53–56]. Indeed, as we wanted to collect and access data in real-time and rapidly given the ever-evolving nature of the COVID-19 pandemic, the use of an easily accessible online questionnaire is pertinent. A web-based questionnaire was easy to use for participants as they could access it using any electronic device when most convenient for them, a strategy tailored for young adults, our target population. Additionally, because we wanted to reach as many pregnant persons as possible to ensure as much diversity and representation as possible, the web-based strategy was preferred. To further ensure that our sample would be representative, recruitment was also done in person at the Montreal Dietetics Dispensary, which provides support to low-income and newly arriving mothers.
In China, recruitment was done in person in three central hospitals from the Henan Province by investigators when pregnant persons came to the hospital for their clinical follow-up and were handed an electronic device where they completed the same questionnaire online independently. The difference in recruitment strategy was due to restricted social media access in China [57].”
Finally, because we wanted to evaluate the direct impact of different COVID-19 related public health measures on maternal mental health, the use of the same validated and standardized instruments in both countries was essential. As such, we chose to use the Edinburgh Perinatal Depression Scale (EDPS) and the Generalized Anxiety Disorders scale (GAD-7) because these tools are widely used in the evaluation of maternal mental health, are validated in Mandarin, French and English, and are reliable to screen depression and anxiety. We have also described the reason why the definition cut-offs were used. This section of the methods, on pages 10-11, lines 304-323 reads as follows:
“E) We lastly assessed maternal mental health during the COVID-19 pandemic by measuring: 1) Maternal depression during the pandemic, using the Edinburgh Perinatal Depression Scale (EPDS)[58], 2) Anxiety during the pandemic, using the generalized anxiety disorders scale (GAD-7) [59], 3) Satisfaction with life, comparing the time prior to vs. since the start of the COVID-19 pandemic using a 4-category ordinal scale with responses ranging from “very satisfied” to “very unsatisfied” and 4) Stress due to this COVID-19 pandemic using a visual analog scale ranging from 0 (no stress) to 10 (maximum stress). We chose those the EDPS and GAD-7 instruments to assess maternal mental health. The EDPS score has been validated in Mandarin, French and English [60–62]. This instrument is composed of 10 items. Each item poses a question and is scored from 0 to 3, and the total scores range from 0 to 30. With a cut-off value of ≥13 representing severe depression, this tool has a sensibility of 66% and a specificity of 95% for the screening of depression[63]. The GAD-7 scale has also been validated in Mandarin, English and French [64–66]. This instrument is comprised of 7 items, each item poses a question and is scored from 0 to 3 and the total score ranges from 0 to 21. With a cut-off value >9 representing moderate to severe anxiety, this score has a sensibility of 89% and a specificity of 82% for the screening of anxiety [67]. As such, we have categorized depression symptoms as continuous measure first and further classified as moderate to severe (if EPDS >9) and severe (if EPDS ≥13) [22]. Similar to this, anxiety symptoms were classified as moderate to severe (if GAD-7 >9), and severe (if GAD >15)[23]. These cut-offs are determined by the tools themselves [22-23]”

Reviewer 2 Report
Authors have addressed my comments. I recommend this article for publication in present form.
Author Response
Reviewer 2
Authors have addressed my comments. I recommend this article for publication in present form.
Response: We thank you for reviewing our comments and for recommending our manuscript for publication.